# A neural progenitor mitotic wave is required for asynchronous axon outgrowth and morphology

Jérôme Lacoste[1]*, Hédi Soula[2], Angélique Burg[1], Agnès Audibert[1], Pénélope Darnat[1], Michel Gho[1]*†, Sophie Louvet-Vallée[1]*†

[1]Sorbonne Université, CNRS, Laboratoire de Biologie du Développement - Institut de Biologie Paris Seine (LBD-IBPS), Cell cycle and cell determination Team, Paris, France; [2]Sorbonne Université, INSERM, NutriOmics Research Unit, Paris, France

**\*For correspondence:**
jerome.lacoste@sorbonne-universite.fr (JL);
michel.gho@sorbonne-universite.fr (MG);
sophie.louvet_vallee@sorbonne-universite.fr (SL-V)

†These authors contributed equally to this work

**Competing interest:** The authors declare that no competing interests exist.

**Abstract** Spatiotemporal mechanisms generating neural diversity are fundamental for understanding neural processes. Here, we investigated how neural diversity arises from neurons coming from identical progenitors. In the dorsal thorax of *Drosophila*, rows of mechanosensory organs originate from the division of sensory organ progenitor (SOPs). We show that in each row of the notum, an anteromedial located central SOP divides first, then neighbouring SOPs divide, and so on. This centrifugal wave of mitoses depends on cell-cell inhibitory interactions mediated by SOP cytoplasmic protrusions and Scabrous, a secreted protein interacting with the Delta/Notch complex. Furthermore, when this mitotic wave was reduced, axonal growth was more synchronous, axonal terminals had a complex branching pattern and fly behaviour was impaired. We show that the temporal order of progenitor divisions influences the birth order of sensory neurons, axon branching and impact on grooming behaviour. These data support the idea that developmental timing controls axon wiring neural diversity.

## Editor's evaluation

This study investigates development of the mechanosensory organ on *Drosophila* notum. It combines live imaging, mathematical modelling, genetics and behavioural analysis to show that, in the peripheral nervous system of *Drosophila*, entry of progenitor cells into mitosis is spatially and temporally controlled. The authors suggest that this ensures proper targeting of sensory neurons within the ventral nerve cord. This timing is important for axonogenesis and proper spatial arborization, ultimately influencing the animal's behaviour. The study will be of broad interest to those who work on the developmental of sense organs, and in general on the role of timing in development.

## Introduction

It is commonly accepted that nervous system development relies on the precise spatio-temporal regulation of gene expression in neural progenitors (*Holguera and Desplan, 2018*). However, little is known about how neural diversity can be generated among neural progenitors that are homogenously specified (*Hirata and Iwai, 2019*), for example, in neurons that connect peripheral sensory organs with the central nervous system (*Gatto et al., 2019*).

Microchætes are peripheral mechanosensory organs on the thorax of *Drosophila melanogaster*. These organs arise from sensory organ progenitor cells (SOPs) which are selected among G2 arrested cells of proneural clusters during pupal stage (*Sato et al., 1999*; *Usui and Kimura, 1993*). In the dorsal region of the thorax (notum), SOPs appear sequentially from five parallel proneural rows: first rows

5, followed by rows 1 and 3 and finally rows 2 and 4 (*Usui and Kimura, 1993*; *Corson et al., 2017*). SOPs selection involves Notch-dependent lateral inhibition from 6 to 12 hr after pupal formation (APF). Then, at 16.5 hr APF, SOPs resume the cell cycle and divide to generate a posterior secondary precursor cell (pIIa) and an anterior secondary precursor cell (pIIb). After subsequent asymmetric cell divisions, they give rise to the external (the socket and the shaft cells) and internal (the neuron, the sheath, and the glial cell) cells respectively (*Gho et al., 1999*; *Fichelson and Gho, 2003*). During these asymmetric cell divisions, daughter cells acquire different cell fates relied with a differential activation of the Notch (N) pathway (*Schweisguth, 2015*). During organogenesis, the bipolar neurons project their dendrite toward the base of the bristle, and their axons toward the ventral thoracic ganglion. The axons enter the ganglion through the posterior dorsal mesothoracic nerve root and extend branches anteriorly and posteriorly (*Ghysen, 1980*). The resulting arborization of the microchæte axon projections is variable. Their complexity is not correlated to the position of the organ but to the time at which rows appear. Thus, the microchætes form the earlier born rows are more branched and arborised over a greater area within the neuropil than those from later born rows (*Ghysen, 1980*; *Usui-Ishihara and Simpson, 2005*). In adult flies, mechanical stimulation of thoracic bristles induces a cleaning reflex in which the first or third leg sweeps the stimulated area. This reflex requires the correct connectivity of the bristle neurons (*Corfas and Dudai, 1989*; *Ghysen, 1980*).

SOPs distribution in the notum depends on N-signaling pathway as well as modulators, such as Scabrous (Sca). Scabrous is a secreted fibrinogen like protein that interacts physically with N and Dl and modulates their activities (*Mok et al., 2005*), The precise mechanism by which Sca regulates the N-signalling is unknown and moreover, is tissue specific. Thus, in the notum, *sca* mutants have an excess of bristles, a phenocopy of *N* mutants, indicating that Sca positively modulates N-activity (*Renaud and Simpson, 2001*). In wing discs, ectopic expression of *sca* inhibits N-pathway by reducing the interactions with its ligands (*Lee et al., 2000*). In eye discs, some studies indicate that Sca promotes N activation in response to Dl (*Li et al., 2003*) and others show that expression of *sca* in receptor precursor cells reduces N-activity (*Munoz-Soriano et al., 2016*; *Powell et al., 2001*).

In this work, we investigated how resumption of SOP division within the notum influences neurogenesis and ultimately fly behaviour. In a kinetic study, we show that SOPs enter mitosis successively according to a temporal wave that propagates from the centre towards the anterior and the posterior border of the thorax. We took a genetic approach to demonstrate that the Notch-ligand Delta and the secreted Notch-signalling modulator Scabrous control this wave. By altering cellular morphology, we show that propagation of the wave is mediated by cell-cell interactions through cell protrusions from SOPs. Finally, we present evidence showing that this mitotic wave has an impact on neural wiring and cleaning behaviour. Overall, our data support the idea that timing of neural progenitor divisions controls normal wiring in the central nervous system.

## Results

### SOP mitosis resumes in a temporal wave

To monitor mitosis in SOPs, transgenic pupae at 15 hr APF expressing GFP under the SOP specific *neuralised* driver (*neurD> GFP*) were monitored with live imaging. As a way to comparing nota, the first cell in each row to divide was used as both a temporal (division time, min) and a positional reference (metric rank, μm), designated $SOP_0$. Centring our attention on SOPs located in the dorsal most region of the thorax (Rows 1–3 in *Figure 1A*), we observed that the $SOP_0$ cells were located in the anteromedial region of each row and generally, the more distant a SOP was from its corresponding $SOP_0$, the later it divided (*Figure 1A–B*, *Figure 1—figure supplement 1A* and *Video 1*). The order of mitosis of SOPs was not strict. However, plotting the time of SOP division against its metric rank relative to $SOP_0$, reveals that on average SOPs resumed the cell cycle in a wave of mitoses spreading towards the anterior and the posterior ends along each row. Thus, all SOPs in a row divided in two hours for all rows analysed (*Figure 1B–C*). No statistical difference was found between the rate of the mitotic waves in rows 1, 2, and 3 (p = 0.3954, ANOVA) or between left and right rows (p = 0.3038, ANOVA). $SOP_0$ were not located precisely in the middle of each row, so the number of SOPs in the anterior and posterior portion of each row was unequal (*Figure 1—figure supplement 1*, *Figure 1—figure supplement 2*). Still, no statistical difference was found between the rates of the anterior and posterior mitotic waves (p = 0.0689, ANOVA). To characterise each curve, the rate of the mitotic wave

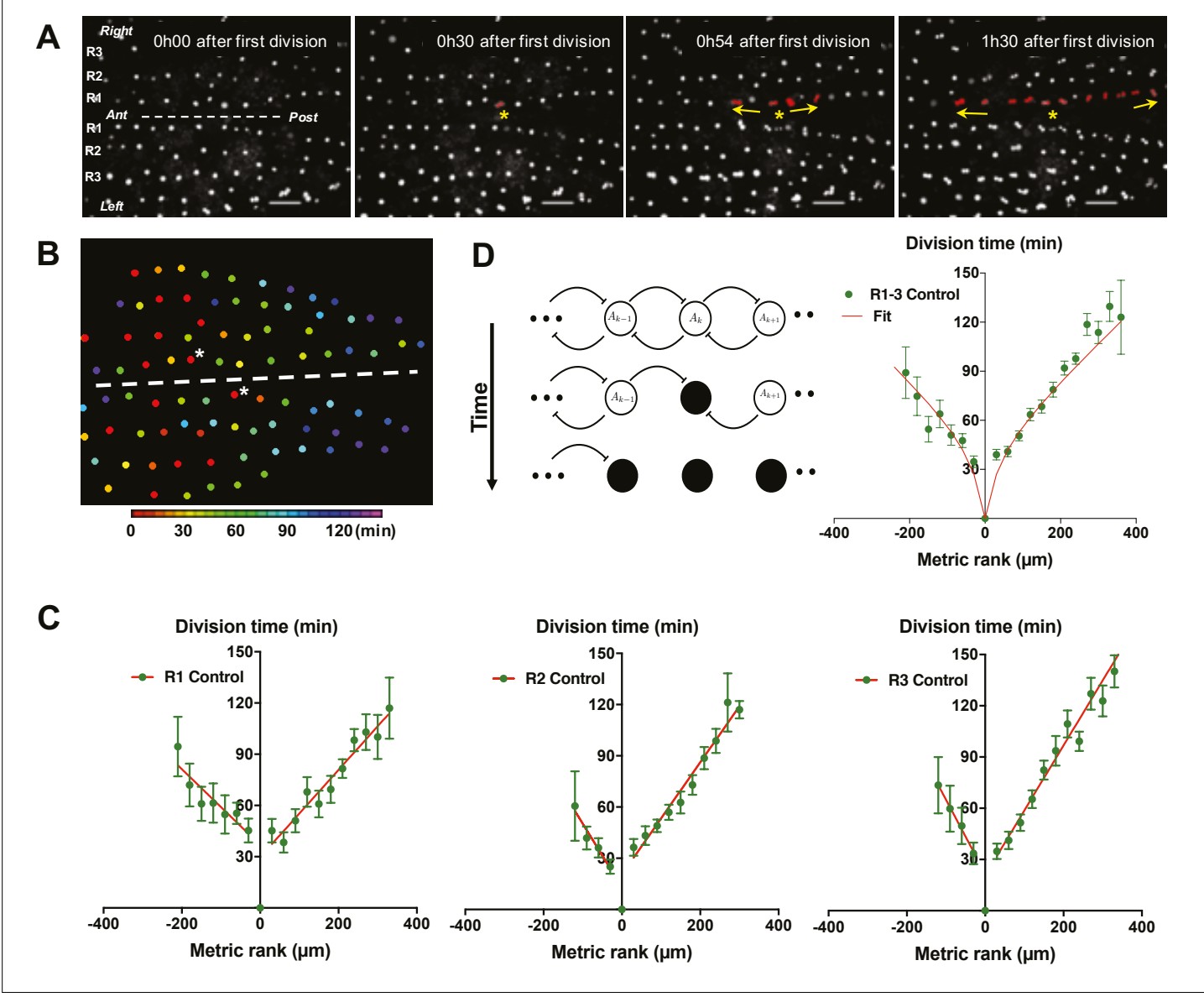

**Figure 1.** G2-arrested SOP cells of the neuroepithelium resume division in a temporal wave. (**A**) Four frames of a time-lapse recording of a control *Drosophila melanogaster* pupa from *Video 1*. The first three rows of cells in the most dorsal part of the thorax are indicated as R1, R2, and R3 respectively. The dashed line marks the midline, with anterior (Ant) to the left and posterior (Post) to the right. SOPs that have divided are highlighted in red in R1. The yellow arrows indicate the propagation of the wave from SOP$_0$ marked by a yellow star. Scale bar, 50 μm. (**B**) Heatmap of a representative control notum depicting time of SOP division. Circles represent SOP cells arranged in rows. The relative time of division is colour coded according to the scale, each colour covering 6 min. Only rows 1–4 are represented. The dashed line shows the midline, with anterior to the left, and SOP$_0$ in row one marked by white stars. (**C**) The time of SOP cell division (mean time± SEM of n = 16 nota) in row 1 (**R1**), row 2 (**R2**), and row 3 (**R3**) is plotted according to position both relative to the position and time of division of the first cell to divide in each row. Negative and positive metric rank therefore corresponds to SOPs that are anterior and posterior to the SOP$_0$, respectively. Red lines show standard linear regressions. SOPs were identified by GFP expression in a *neur> GFP* fly line. (**D**) Modelling the wave of SOP divisions. Left, a schematic view of the model at three consecutive times. Empty circles depict non-dividing cells. The flat-headed arrows indicate the inhibition that one cell exerts on immediate neighbours preventing entry into mitosis. Filled circles represent mitotic or post-mitotic cells which no longer have an inhibitory effect thus allowing neighbours to divide. These interactions allow the progression of mitosis along a row. Right, fit of experimental data with theoretical values obtained with the model using as parameters $\rho$ = 20 and μ = 0.65 (See Model description).

The online version of this article includes the following source data and figure supplement(s) for figure 1:

**Source data 1.** Related to *Figure 1* Raw data.

**Figure supplement 1.** Illustrations of individual mitotic waves in control and *sca^BP2^* mutant.

*Figure 1 continued on next page*

*Figure 1 continued*

**Figure supplement 1—source data 1.** Related to *Figure 1—figure supplement 1*.

**Figure supplement 2.** The mitotic wave origin spreads out in several genetic contexts.

**Figure supplement 2—source data 1.** Related to *Figure 1—figure supplement 2* Raw data.

**Figure supplement 3.** Reduced mitotic waves were better fitted with low values of the inhibitory parameter μ.

**Figure supplement 3—source data 1.** Related to *Figure 1—figure supplement 3* Raw data.

was calculated as the mean absolute value of the inverse of the slopes of the linear approximation of the curves for all conditions. For control conditions, this gives for the mean rate 3.14 µm min$^{-1}$ (see *Figure 1* and ). We conclude that the resumption of SOP mitosis is not random because the first division is in a specific region of the notum and a steady wave of division propagates towards the anterior and the posterior parts of the notum.

To establish working hypotheses on the mechanism involved in the propagation of this mitotic wave, we developed a simple contagion model assuming that (1) neighbouring progenitor cells in a row inhibit each other from entering mitosis through cell-cell contact, (2) this inhibition is switched off when cells divide, (3) mitoses occur whenever a cellular compound concentration reaches a certain threshold and (4) this compound is produced at a constant rate. Given these assumptions, the system can be modelled by two differential equations characterised by four parameters (see Appendix 1). These parameters are: r, the rate of production of a pro-mitotic factor A; µ, the rate of production of I, which inhibits A; $\rho$ and δ are degradations times of A and I, respectively. With this set of differential equations, our model accounts for the observed propagation of SOP mitoses very well (*Figure 1D*).

In particular, this simple model accounts for the non-linearity of the curve observed. Indeed, the interplay between the progressive capacity of SOPs to divide (modelled by the factor A) and the inhibitory effect on mitotic entry due to neighbouring cells (modelled by the factor I that inhibits A's production) brings cells farther from the start closer and closer to threshold to divide. The result is that, proportionally, cells take less and less time as the mitotic wave progresses along the row. Therefore, the mitotic wave accelerates over time. In other words, the relation between division time and SOP position is flattened for SOP position values far from the origin. This flattening is particularly observed for low inhibition conditions (*Appendix 1—figure 1*). Conversely, at a very strong level of inhibition, the resulting curve is almost linear with a slope different to zero (*Appendix 1—figure 1*). The non-linearity of the curve is accentuated between the 1st and 2nd division by the fact that the point 0, the location of SOP$_0$, is also the temporal origin. As such, the time of division of SOP$_0$ is for definition 0 and it is not a mean as the other points. Since all other values are positive (all cells divide obviously after the SOP$_0$), this induces a positive jump in the delay between the 1st and following divisions.

Using this model, variations of only two parameters, $\rho$ and µ, account for our experimental data (See Parameters estimation in *Appendix 1—figure 2*). In the control condition, the best fit was obtained with values of $\rho$ and µ of 40.38 and 0.52, respectively (*Figure 1—figure supplement 3—source data 1*).

## Rac1$^{N17}$ affects both cytoplasmic protrusions length and SOP mitotic wave progression

Based on this model, SOPs are likely to influence neighbouring SOPs, so we analysed the potential ways by which they interact. SOPs in each row are actually separated by three to four epithelial cells, but SOPs produce dynamic actin-based protrusions several cell diameters in length that physically interact with protrusions from neighbouring

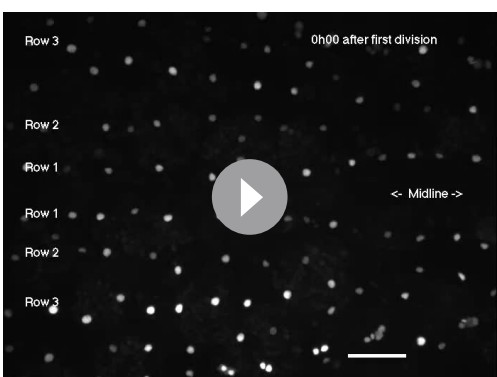

**Video 1.** Time lapse recording of a control pupa during a 4 hr period beginning at 15 hr after pupal formation. The wave of SOP division is highlight in red in the row 1. Each frame was obtained by combining a z-stack (composed of 10 optical sections separated by 2 µm) acquired every 3 min. During the in vivo imaging, the pupae was maintained at 30 °C. Anterior is to the left and view is dorsal. Scale bar represents 50 µm.
https://elifesciences.org/articles/75746/figures#video1

SOPs and epithelial cells (*Figure 2A* left) (*Cohen et al., 2010*; *Hunter et al., 2016*; *Renaud and Simpson, 2001*). When SOPs divide, they become spherical and protrusions are drastically reduced (*Figure 2—video 1*). To test the possibility that these protrusions control the mitotic wave, we overexpressed a dominant negative form of Rac1 (Rac1[N17]) (*Figure 2A* right and *Figure 2—video 2*). Rac1, a factor implicated in F-actin polymerisation, is required for the growth of protrusions in SOPs (*Cohen et al., 2010*). To estimate the protrusion length, we have measured the protrusion extension area (*Figure 2B*). Under these conditions, SOP protrusions were shorter than in the control (*Figure 2C*, $p < 0.0001$, ANOVA). Although SOP divisions started at similar developmental times (control 15.7 ± 0.5 hr APF, n = 9; *rac1[N17]* overexpression 15.9 ± 0.5 hr APF, n = 6), we observed that SOPs with shorter protrusions divide more synchronously than in the control (*Figure 2E*, mitotic wave rate 6.95 µm min$^{-1}$). Indeed, all SOPs in a row divided within one hour for all rows analysed (*Figure 2E*, Row 1 $p < 0.0001$; Row 2 $p = 0.0292$, Row 3 $p = 0.0014$, ANOVA, and *Figure 2—figure supplement 1B*). The lesser slopes of the time-rank curves of the mitotic wave in flies with short SOP protrusions shows that neighbouring SOPs resume mitosis earlier than normal. Although we cannot exclude that the effect observed with *rac1[N17]* is exclusively due to the diminution of protrusions size, the observation that the protrusions regress prior to the SOP division, leads us to hypothesise that SOPs exchange inhibitory signals throughout protrusions to maintain neighbouring SOPs in G2-arrest. When a SOP divides, its protrusions are retracted so the inhibitory signal is suppressed allowing progression of the mitotic wave to the next SOP along the row. Interestingly, after fitting the experimental data with the model, the best fit was obtained with no change in the $\rho$ value (in row 1, 40,38 in control and 39,12 in *rac1[N17]*) whereas the value of the inhibitory parameter (µ) decreases (in row 1, 0,52 in control and 0,36 in *rac1[N17]*). For the other rows, the parameter values for the best fit are shown in *Figure 1—figure supplement 3B*.

## The SOP mitotic wave is regulated by scabrous and delta

Cellular protrusions allow molecules to be exchanged between cells for direct and selective signalling (*Buszczak et al., 2016*; *Kornberg and Roy, 2014*). The candidate for such a signal would be a secreted molecule with a role in regulating the interactions and spacing of cells in the nervous system. One such molecule is the fibrinogen-like protein Scabrous (Sca) that regulates several processes such as the regularly spaced pattern of ommatidia in the eye or of bristles in the notum (*Baker et al., 1990*; *Cohen et al., 2010*; *Gavish et al., 2016*; *Renaud and Simpson, 2001*), or ommatidial rotation (*Chou and Chien, 2002*). In the protein trap line *sca::GFP*, Sca was detected as multiple spots of fluorescence in the cytoplasm and along the protrusions of SOPs (*Figure 2F* and *Figure 2—video 3*). At 14h30 APF, very faint Sca spots were detected on the notum. The intensity of these spots reaches a peak at 24 hr APF then decreases rapidly and the spots were undetectable at 32 hr APF (*Figure 2—figure supplement 2*). To investigate whether Sca is involved in the synchronisation of SOP mitosis, we studied the mitotic wave in the *sca[BP2]* null mutant, which is viable at the developmental period studied. In this mutant, we observed SOP$_0$ cells at different points along each row as they were not limited to the anterior region (*Figure 3A*, *Figure 1—figure supplement 1B* and *Figure 1—figure supplement 2B, C*). A similar effect on the localisation of SOP$_0$ was observed after overexpression of *rac1[N17]* (*Figure 1—figure supplement 2D-E*). More importantly, in the *sca[BP2]* mutant, SOPs divided more simultaneously than in the control (Row 1, $p = 0.0015$; Row 2, $p = 0.0029$; Row 3, $p = 0.0014$, ANOVA, *Figure 3A–B*, *Figure 1—figure supplement 1*, *Figure 3—figure supplement 2A*, *Video 2*). For instance, Row one in *sca[BP2]* divided in one-third of time taken for the control SOP row to divide, and the rate of the wave increased to 20.27 µm min$^1$ (See *Figure 1* and ). This change was not associated with change in the cell fate of the sensory organs (*Figure 3—figure supplement 1*). The release of the inhibition observed in *sca[BP2]* context was again well confirmed by fitting the experimental data with the model. For instance, for row 1, the best fit was obtained with no change in the $\rho$ value (37,71 in control and 41,08 in *sca[BP2]*), whereas the value of the inhibitory parameter (µ) decreases from 0,52 (control) to 0,31 (*sca[BP2]*). The invariance of the $\rho$ and the reduction of µ values were observed for all rows (*Figure 3C*, *Figure 1—figure supplement 3E-F* and *Figure 1—figure supplement 3—source data 1*). This Sca-mediated inhibition is not likely to be related to protrusion length shortening because no difference was found between *sca[BP2]* and control SOP protrusions (*Figure 2D* and see *Renaud and Simpson, 2001*). To exclude the possibility that the absence of Sca during the entirely development of *sca[BP2]* mutant could secondarily impact the mitotic wave, *sca*

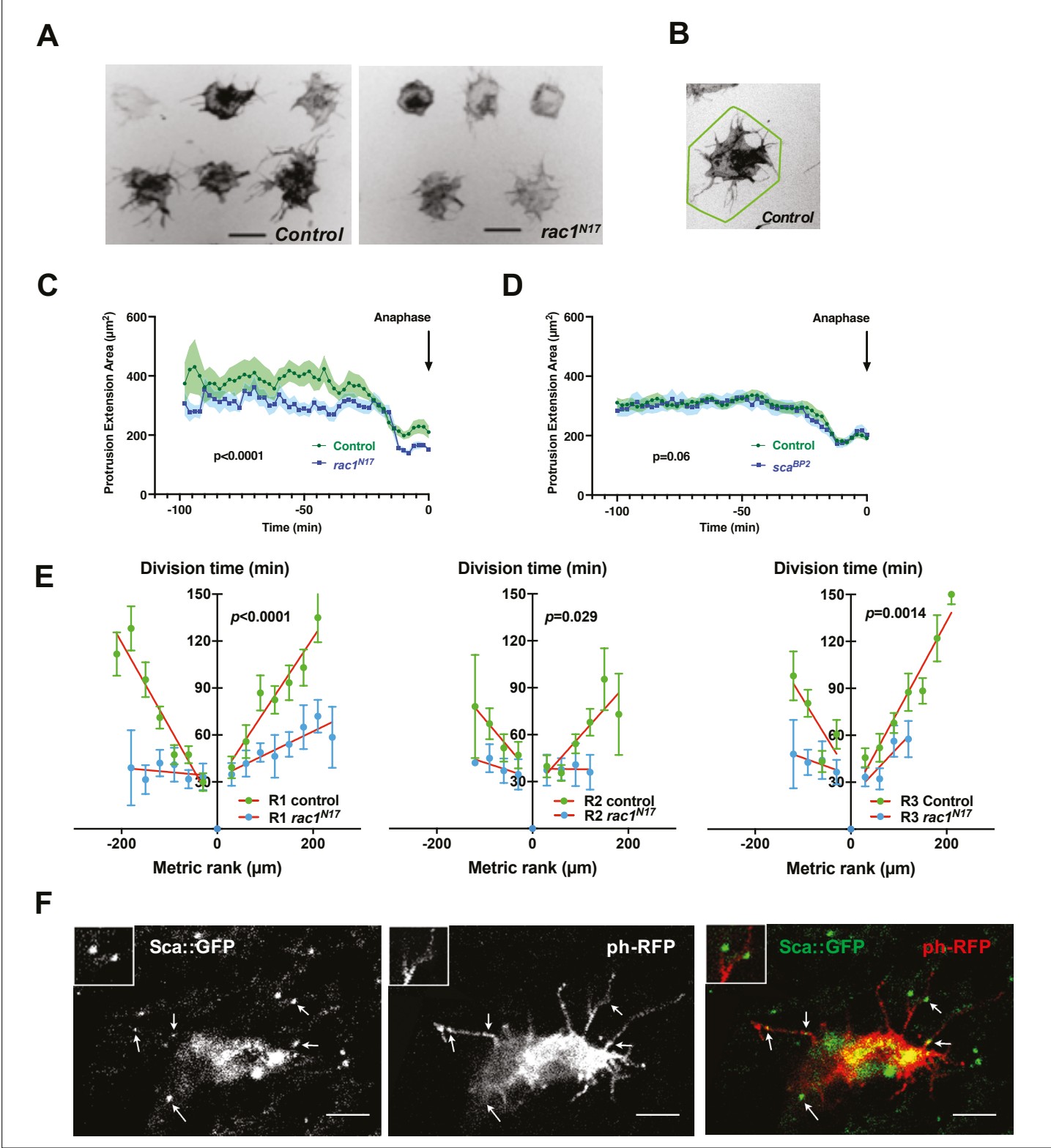

**Figure 2.** The SOP mitotic wave is affected in conditions where cell protrusions are reduced. (**A**) Frames from live recordings showing cell protrusions of SOPs in control (left) and *rac1^N17* (right) pupae. *tubGal80^ts* was used to conditionally express *Gal4* and thus *rac1^N17* by maintaining flies at 18 °C and shifted them to 30 °C from 12 hr to 19 hr APF. Protrusions were visualised by specific expression of a membrane-tethered RFP form (*ph-RFP*), invert fluorescence. Scale bar, 20 μm. (**B**) Illustration of the polygonal line corresponding to the convex hull enclosing the extremities of the longer protrusions (Fiji software) to quantify the cell protrusion length. (**C**) Protrusion extension area over time in control and after *rac1^N17* conditional overexpression

*Figure 2 continued on next page*

*Figure 2 continued*

(mean ± SEM, n = 11 cells for control and n = 14 for *rac1^N17*) measured from live recordings. Anaphase was taken as a temporal reference. (**D**) Protrusion extension area over time in control and in *sca^BP2* null mutant (mean ± SEM, n = 20 cells for control and n = 18 for *sca^BP2*) measured from live recordings. Anaphase was taken as a temporal reference. (**E**) The SOP mitotic wave in row 1 (**R1**), row 2 (**R2**) and row 3 (**R3**) in control and after overexpression of *rac1^N17*. Mean division time± SEM of n = 6 nota. (**F**) Sca localisation in a SOP cell. Arrows indicate distinct Sca foci along cell protrusions. Scabrous (left panel) in SOP cell marked with a membrane-bound ph-RFP (middle panel), merged in right panel. Images are a maximum projection of three confocal sections. Two Sca foci are magnified in the inserts. Scale bar, 5 µm.

The online version of this article includes the following video, source data, and figure supplement(s) for figure 2:

**Source data 1.** Related to *Figure 2* Raw data.

**Figure supplement 1.** Illustrations of individual mitotic waves after reduction of cell protrusions by overexpression of Rac1^N17 and after conditional inactivation of *sca*.

**Figure supplement 1—source data 1.** Related to *Figure 2—figure supplement 1* Raw data.

**Figure supplement 2.** Sca is transiently expressed in the notum.

**Figure 2—video 1.** Time lapse recording showing the dynamic of protrusions in SOP cells in a control pupa.
https://elifesciences.org/articles/75746/figures#fig2video1

**Figure 2—video 2.** Time lapse recording showing the dynamics of protrusions of SOPs in pupa overexpressing the negative form of Rac1 (*rac1^N17*).
https://elifesciences.org/articles/75746/figures#fig2video2

**Figure 2—video 3.** Time lapse showing the dynamics of Sca::GFSTF (in green) in a SOP protrusion visualised using a membrane tethered RFP (in red).
https://elifesciences.org/articles/75746/figures#fig2video3

expression was specifically downregulated using a conditional RNAi strategy between 12 and 19 hr APF (*Figure 3—figure supplement 2B*), that is from just before and during the mitotic wave. In these conditions, as in null mutant, we again observed that the mitotic wave was flattened (*Figure 2—figure supplement 1C*; *Figure 3—figure supplement 2C*; Row 1, p < 0.0001; Row 2, p = 0.69; Row 3, p = 0.003, ANOVA). In this context, we also observed $SOP_0$ location was not limited to the anteromedial region (*Figure 1—figure supplement 2D, F*). Again, this effect was best fitted by a reduction on the inhibitory parameter compared to the control (*Figure 1—figure supplement 3A, C and F* and *Figure 1—figure supplement 3—source data 1*). These data indicate that Sca transported through cell protrusions specifically controls the wave of SOP mitoses along the rows.

Sca is known to modulate the Delta/Notch (Dl/N) pathway (*Petruccelli et al., 2018*), so we tested whether this pathway is involved in regulating the SOP mitotic wave. Since null *Dl* mutants are lethal, we studied the mitotic wave in *Dl^7/+* heterozygous background. We did not observe a statistically significant reduction in the mitotic wave rate in *Dl^7/+* SOPs (*Figure 3D*, p = 0.69, ANOVA). Similarly, we did not observe any difference in the rate of the SOP division wave in the *sca^BP2/+* heterozygous line (*Figure 3E*, p = 0.71, ANOVA). By contrast, when the gene dosage of both *sca* and *Dl* was reduced by half, the time-rank curves were flattened corresponding to a significant increase in the mitotic wave rate (from 3.14 to 43.35 µm min⁻1) (*Figure 3F*, p = 0.048, ANOVA). Again, this effect was best fitted by a reduction on the inhibitory parameter (*Figure 3G*, *Figure 3—figure supplement 3* and *Figure 1—figure supplement 3—source data 1*). These results demonstrate that *sca* and *Dl* genetically interact, which suggests that these factors act in concert to control the SOP mitotic wave. Indeed, we observed that Dl and Sca co-localise in vesicle-like structures inside the cytoplasm with Dl surrounding the Sca staining (*Figure 3H*). This co-localisation firstly described by *Renaud and Simpson, 2001*, evidences that these two proteins are transported from the endoplasmic reticulum to the membrane in the same vesicles.

Altogether, our results show that the wave of SOP mitoses along rows depends on an inhibitory signal transmitted by protrusions from the progenitors and is controlled by Dl and Sca proteins.

## Outcome of the mitotic wave on microchæte axonogenesis

Considering the invariability of sensory organ formation, we wondered whether differences in the wave of SOP mitosis might impact the timing of axonogenesis along each row. To this end, we measured axon length at the beginning of axonogenesis (24 hr APF) in control and *sca^BP2* pupae (*Figure 4A*). In control pupae, the axon length varied along the antero-posterior axis. Indeed, within a row, more the neurons were located at the extremities, the shorter their own axon. However, in *sca^BP2* animals, the length of the axon along a row was significantly more homogeneous (*Figure 4B*, p = 0.0232, ANOVA).

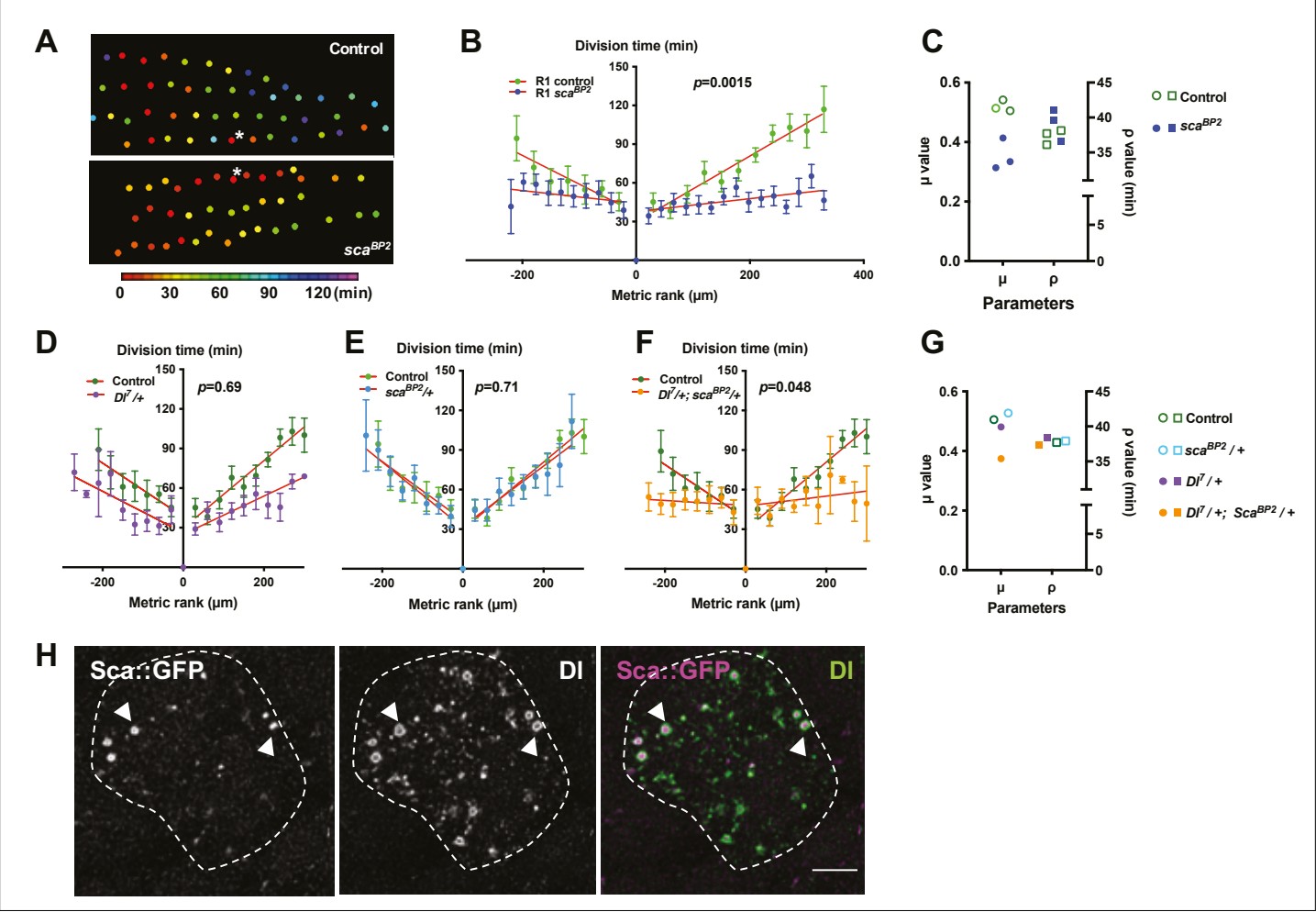

**Figure 3.** The mitotic wave of SOP cells is controlled by Notch/Delta/Scabrous signals. (**A**) Heatmap of a representative heminota in control and *sca^BP2* homozygous null mutant aligned, anterior to the left. The relative time of SOPs division is colour coded according to the scale. (**B**) The SOP mitotic wave in row one in control and in *sca^BP2* homozygous mutant. (**C**) Dot plot showing the distribution of μ and ρ parameters extracted from the best fits displayed in *Figure 1—figure supplement 3D, E*. Note that the best fit was obtained in *sca^BP2* null mutant after reduction of the μ parameter only. (**D–F**) Genetic interactions between *Dl^7* and *sca^BP2*. Comparison of the SOP mitotic wave in control and *Dl^7/+* (**D**), *sca^BP2/+* (**E**) and *Dl^7/+, sca^BP2/+* (**F**). Mean division time± SEM, n = 9, 8, and 11 nota respectively. (**G**) Dot plot showing the distribution of μ and ρ parameters extracted from the fits displayed in *Figure 1—figure supplement 3D, E*. . Note that in simple heterozygous backgrounds, μ and ρ were similar to the control while in *Dl^7/+, sca^BP2/+* double heterozygous background, the best fit was obtained after reduction of the μ parameter only. (**H**) Confocal image of one SOP in a sca::GFSTF protein-trap fly at the moment of the SOP wave (16 hr APF), immunostained for GFP (left panel) and Dl (middle panel), merged in the right panel (purple and green respectively). Arrowheads show co-localisation of Dl and Sca into vesicle-like structures with Sca::GFSTF as puncta surrounded by Dl staining. SOP soma is delimited by a dashed line. Scale bar, 5 μm.

The online version of this article includes the following source data and figure supplement(s) for figure 3:

**Source data 1.** Related to *Figure 3* Raw data.

**Figure supplement 1.** The fate of sensory organ cells is not affected in *sca* mutant.

**Figure supplement 2.** The amplitude of the mitotic wave is reduced in *sca* loss of function.

**Figure supplement 2—source data 1.** Related to *Figure 3—figure supplement 2* Raw data.

**Figure supplement 3.** The best fit of the mitotic wave with a reduced amplitude is obtained after preferential lowering the inhibitory parameter μ.

**Figure supplement 3—source data 1.** Related to *Figure 3—figure supplement 3* Raw data.

These observations indicate that as a result of the SOP mitotic wave, neurogenesis and consequently axonogenesis progress from the centrally located neurons towards those located at the extremities in controls, while in *sca* mutants, in which the wave was disrupted, axonogenesis occurs simultaneously along each row. Thus, the mitotic wave of SOP division controls axonogenesis timing.

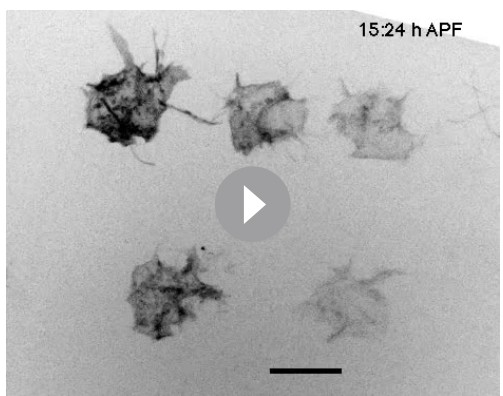

15:24 h APF

**Video 2.** Time lapse recording of the SOP mitotic wave in control pupa (top panel) and in sca^BP2 mutant pupa (bottom panel). For comparison, two hemi-nota are aligned. The observed difference in SOP nuclei size in the two movies is due to the lineage markers used (nuclear GFP and histone-RFP). The waves of SOP divisions are highlighted in red in the rows 1. Anterior is to the left and view is dorsal. Scale bar represents 50 μm.

https://elifesciences.org/articles/75746/figures#video2

## Disruption of the SOP mitotic wave leads to a modified organisation of microchæte axonal projections

We wondered whether the altered axonogenesis timing observed in the mutant bristle organs influences the patterning of axonal endings in the thoracic ganglion. To address this question, the morphology of these axon terminals was studied by expressing a membrane-bound GFP form in bristle cells using *pannier-GAL4*, as a driver of dorsal expression. Under these conditions, the neuropile in the thoracic ganglion formed by axon terminals from all bristles of the central dorsal notum was visualised as a mesh-like triangular structure formed by an anterior arm at each left and right sides and a posterior commissure that cross the midline (*Figure 4* C1). In 11% of ganglia analysed in control flies a second medial commissure was observed (*Figure 4—figure supplement 1* arrow). Interestingly, in flies in which the mitotic wave was disrupted after *sca* downregulation (*RNAi-sca*), this percentage increased to 34% (*Figure 4—figure supplement 1*, p = 0.0002, Fisher's exact test). These observations prompted us to analyse single axon terminals by immunodetecting individual axons using the stochastic labelling method MCFO (*Nern et al., 2015*). We observed in control flies that individual axon terminals have two main primary branches, an anterior branch that does not cross the midline and a posterior branch that crosses the midline (*Figure 4* C1 and C3 top panel). In *RNAi-sca* animals, we observed that axon terminals are more branched (*Figure 4* C3 bottom panel). The lengths of primary anterior and posterior branches, between the root and the first branch point (*Figure 4* C2), of *RNAi-sca* axons were significantly shorter than those of control axons (*Figure 4D*, p < 0.0001 and p = 0.0001 respectively, ANOVA). Since on the one hand, *sca* was inactivated specifically and only during the mitotic wave, and on the other hand, *sca* expression turn-off prior to axonogenesis (*Figure 2—figure supplement 2*), the effects observed could not be due to a possible effect of *sca* on axonogenesis itself. As primary branches were shorter when SOP division occurred simultaneously on the notum, the timing of axonogenesis is likely to be an important factor controlling axon branching. Interestingly, our data are in agreement with studies showing that axon terminals of bristles from rows that are formed earlier, are more branched than those from rows that develop later (*Usui-Ishihara and Simpson, 2005*). As such, our results led us to hypothesise that the order of birth of neurons in a given row contributes to the future connectivity of these neurons. Further experiments aimed to analyse the axon terminal of sensory organs at a given row are required to confirm this hypothesis.

## Disruption of the SOP mitotic wave leads to changes in fly behaviour

Finally, we wondered whether modifications in axon branching after disruption of normal timing of SOP mitosis led to behavioural changes. In order to test this possibility, we analysed the cleaning reflex, as a read out for the function of the mechanosensory system of flies, in conditions where the SOP mitotic wave was disrupted. The cleaning reflex is a patterned set of leg movements elicited in a fly when its thoracic bristles receive tactile stimulation (*Figure 5A* and *Figure 5—video 1*; *Corfas and Dudai, 1989*; *Vandervorst and Ghysen, 1980*). When the SOP mitotic wave was impaired, after specific downregulation of *sca* during the wave, we found that the number of air puffs required to elicit a leg response was significantly higher than in control flies (*Figure 5B*, p = 0.028, ANOVA).

Altogether, our results show that the temporal order of neural precursor cells division impact on neuronal cell diversity and ultimately on behaviour.

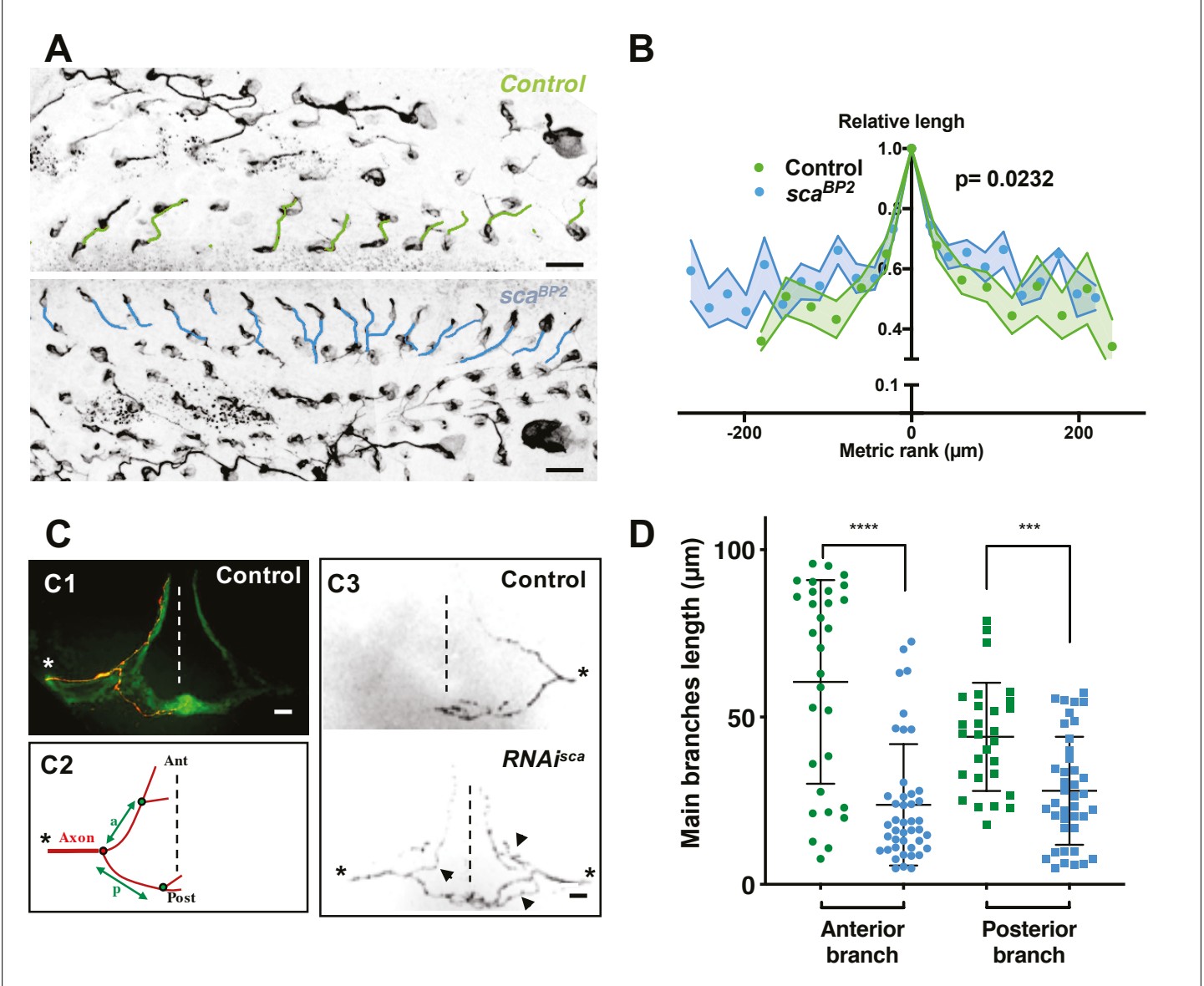

**Figure 4.** Microchæte axonal projections are affected after SOP mitotic wave disruption. (**A**) Axon immunostaining using anti-Futsch/22C10 antibodies in control (*w1118*) and *scaBP2* heminota at 24 hr APF aligned, anterior to the left. Axons in row one were artificially coloured in green for *w1118* and blue for *scaBP2* homozygous flies. (**B**) Relative axon length of neurons belonging to sensory organs on row one plotted against relative position (mean length± SEM of n = 18 rows). The neuron with the longest axon on the row was taken as temporal and spatial reference in control (*w1118*) and *scaBP2* homozygous pupae at 24 hr APF. (**C**) Axon projections of single sensory organ neurons in an adult thoracic ganglion identified with MCFO strategy. The midline is indicated by vertical dotted lines, anterior (Ant) up and posterior (Post) down. The asterisks indicate the point of entry of the sensory nerve into the thoracic ganglion. (**C1**) A single axon projection (red) counterstained with the neuropile formed by all axon terminals from sensory organs on the central notum (green). (**C2**) Schematic drawing of an axon projection. The length of the anterior (**a**) and posterior (**p**) primary branches were used to quantify the degree of branching of axon terminals. (**C3**) Representative examples of axon projections into the thoracic ganglion in control fly (top panel) and when the SOP mitotic wave was impaired (*UAS-RNAi-sca*, *tubGal80ts* was used to conditionally express *Gal4* by maintaining flies at 18 °C and shifted them to 30 °C from 12 hr to 19 hr APF, bottom panel). Arrowheads point to additional branches in *RNAi-sca*. (**D**) Lengths of anterior and posterior primary branches in single axon terminals of control (green, n = 30) and in flies where the SOP mitotic wave was impaired (conditional *RNAi-sca*) (blue, n = 43). Mean ± SEM ****p < 0.0001, ***p = 0.0001, two-tailed unpaired t-test.

The online version of this article includes the following source data and figure supplement(s) for figure 4:

**Source data 1.** Related to *Figure 4* Raw data.

**Figure supplement 1.** *sca* downregulation modify the sensory neuropile structure.

**Figure supplement 1—source data 1.** Related to *Figure 4—figure supplement 1* Raw data.

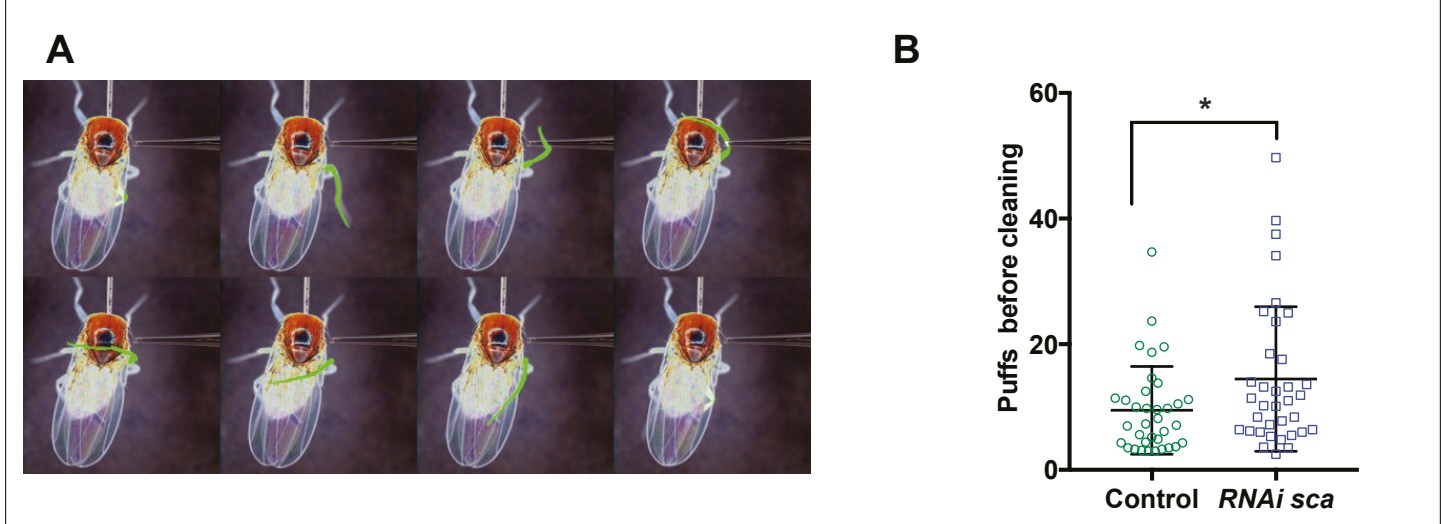

**Figure 5.** Disruption of the SOP mitotic wave leads to changes in fly behaviour. (**A**) Eight frames of a time-lapse recording of a decapitated control fly showing the sequence of movements of the metathoracic leg (artificially coloured in green) upon stimulation of the dorsal bristles. (**B**) Number of air puffs required to elicit a cleaning reflex in control flies (n = 35) and in flies where the SOP wave was impaired (*UAS-RNAi-sca, tubGal80$^{ts}$* was used to conditionally express *Gal4* by maintaining flies at 18 °C and shifted them to 30 °C from 12 hr to 19 hr APF) (n = 35). Mean ± SEM. *p = 0.0238; ANOVA.

The online version of this article includes the following video and source data for figure 5:

**Source data 1.** Related to *Figure 5* Raw data.

**Figure 5—video 1.** Recording of cleaning reflex assay in a decapitated control adult fly.

https://elifesciences.org/articles/75746/figures#fig5video1

## Discussion

To study how functional neuronal diversity can be generated from a homogenous set of neural precursors, we took advantage of the invariant way in which sensory organs are located on the dorsal epithelium of *Drosophila*. This spatial configuration greatly facilitated the study of the relative timing of SOP division and the identification of a distinct temporal wave of SOP mitosis. Asynchrony in mitotic reactivation timing has been described in *Drosophila* larva neuroblasts. This differential timing is related to two cell cycle arrests: one population of neuroblasts is arrested in G2 while another population is arrested in G0 (*Otsuki and Brand, 2018*). G2-arrested neuroblasts resume mitosis earlier than those in G0-arrest. As in our system, it has been proposed that this particular order of division ensures that neurons form appropriate functional wiring. It is relevant that other temporal processes controlling the wiring of peripheral receptors with the central nervous system have been described in the *Drosophila* eye, another highly organised structure (*Fernandes et al., 2017*). It is conceivable that these temporal patterning mechanisms of neurogenesis, to date identified only in organised tissues, could be more widespread.

A core aspect of our work was to link cellular level of complexity (timing of SOP division) with uppermost level (behaviour). In this context, we presented evidences showing that the cleaning reflex was impaired when the SOP mitotic wave was disrupted. The cleaning reflex has been traditionally analysed after stimulation of macrochaetes rather than microchaetes as in the present work. Macro- and microchaetes have different patterns of terminal axon arborisation (*Usui-Ishihara and Simpson, 2005*). As such, it is remarkable that this fly behaviour was significantly affected by altering the timing of microchaete precursor division in the dorsal thorax. We show that the SOP mitotic wave leads to a progressive neurogenesis along each row of microchaetes. This, in turn, would likely induce a particular pattern of microchæte axon arrival in the thoracic ganglion required for the proper organisation of the neuropila in the central nervous system. Although we have documented this progressive axonogenesis, we do not know the strict pattern of axon arrival into the ventral ganglion. It would depend on the order of birth of neurons, and on the geometry of axon projections that fasciculate to form the

dorsal mesothoracic nerves in the ganglion. In any case, we show here that, when *sca* function was specifically downregulated during the SOP mitotic wave, axonogenesis occurs almost simultaneously in each row of microchaetes. This certainly impairs the pattern of axon arrival into the ganglion leading to ectopic axon branching and changes in fly behaviour. It would be interesting to know whether these impairments are specifically due to neurogenesis occurring simultaneously. To test this, we need find a way to induce different patterns of SOP mitotic entry, for instance, a centripetal wave or a random order. If the observed effect is specifically due to the simultaneity, normal behaviour would be expected to be associated with other patterns of SOP division.

We observed that the first SOP to divide ($SOP_0$) was always located in the anteromedial region of each row. This may reflect the existence of a pre-pattern that causes SOPs located in that region to start dividing earlier than the others. Although the anteromedial region corresponds approximately to the posterior limit of expression of the transcription factor BarH1 (*Sato et al., 1999*), no factors specifically expressed in this region have yet been identified. Alternatively, as the location of $SOP_0$ is modified when Sca function was impaired, an interesting possibility is that $SOP_0$ is selected by an emergent process related to cell-cell interaction in the epithelium, rather than by a passive pre-pattern that organises the first events in the notum.

We present evidence indicating that the secreted glycoprotein Scabrous, which is known to interact with the N-pathway to promote neural patterning, controls the kinetics of SOP mitosis in the notum. In proneural clusters, cells that express high levels of Dl and Sca become SOPs, while surrounding epithelial cells activate the N-pathway to prevent acquisition of a neural fate (*Muskavitch, 1994*; *Renaud and Simpson, 2001*; *Buffin and Gho, 2010*). In eye and notum systems, Sca modulates N-activity at a long range. Indeed, during eye development, *sca* is expressed in intermediate clusters in the morphogenic furrow and transported posteriorly in vesicles through cellular protrusions to negatively control ommatidial cluster rotation (*Chou and Chien, 2002*). Similarly, in the notum, SOP protrusions extend beyond several adjacent epithelial cells in which Dl and Scabrous are detected (*Renaud and Simpson, 2001* and this study). Our data show that shorter protrusions (obtained after *rac1^{N17}* overexpression conditions) as well as loss of function of *Dl* or *sca* make the mitotic wave more synchronous. Since, we do not observe a reduction of the global level of *sca* expression associated with the wave progression, it is plausible that Sca, required to maintain SOPs in G2 arrest, is delivered focally through protrusions that are difficult to follow with in vivo analysis. Although we favoured this possibility, we cannot formally rule out the possibility that Rac1^{N17} overexpression affects Sca secretion per se without affects *sca* expression.

As in neuroblasts, G2 arrest in SOP cells is due to the downregulation of the promitotic factor Cdc25/String. Thus, overexpression of *string* in SOPs induces a premature entry into mitosis (*Ayeni et al., 2016*), while overexpression of negative regulators, like Wee1, maintain these cells in arrest (*Fichelson and Gho, 2004*). Possibly Sca negatively regulates *string* expression, perhaps through the N-pathway that it is known to control the level of String (*Deng et al., 2001*; *Krejcí et al., 2009*). Alternatively, it has been recently shown that the insulin-pathway also regulates String level (*Otsuki and Brand, 2018*). Moreover, in muscle precursors, cell proliferation is induced by the insulin-mediated activation of the N-pathway (*Aradhya et al., 2015*). These observations raise the interesting possibility that, in our system, insulin activates the N-pathway and Sca modulates this activation. Further investigations will be required in order to identify the link between Scabrous, the N/Dl- and insulin-pathways in the resumption of mitosis in SOPs.

During nervous system development, the complex patterns of neuronal wiring are achieved through the interaction between neuronal cell surface receptors and their chemoattractive or repulsive ligands present in the environment (*McCormick and Gupton, 2020*). An essential condition for proper axon guidance is the competence of neurons to respond to these environmental clues. It is generally agreed that neuron competence depends on the specific expression of transcriptional factors regulating their identity (*Petrovic and Hummel, 2008*). We show here that the timing of neuron formation is also a factor controlling their terminal morphology. We propose that the SOP mitotic wave induces a particular pattern of arrival of microchæte axons in the thoracic ganglion (*Figure 6*). This pattern establishes a specific framework of guidance cues on which circuits will be built and ultimately influencing an organism's behaviour. Our

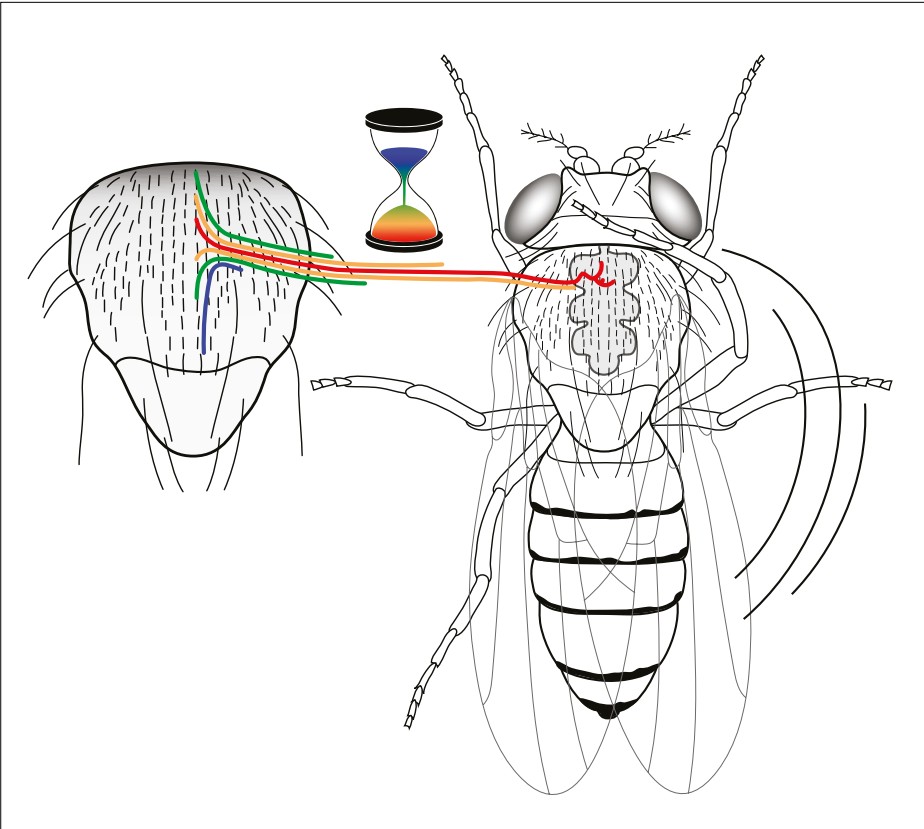

**Figure 6.** A neural progenitor mitotic wave is required for asynchronous axon outgrowth. The mitotic wave of precursor cells induces an asynchronous arrival of sensory axons into the thoracic ganglion. This is depicted here by axons coloured according to the developmental time of sensory organs that in turn is controlled by the wave of precursor cell divisions. Red, early-formed; blue, late-formed sensory organs. This asynchrony would be associated with a particular axon connectivity in the thoracic central ganglion required for a normal cleaning behaviour in the fly.

findings support the idea that, in addition to genetic factors, neurogenic timing is a parameter of development in the mechanisms controlling neural branching.

## Materials and methods

### Key resources table

| Reagent type (species) or resource | Designation | Source or reference | Identifiers | Additional information |
|---|---|---|---|---|
| Antibody | Anti-Futsch (clone 22C10, mouse monoclonal) | DSHB | AB_528403 | IF (1:200) |
| Antibody | Anti-HA (clone 3F10, rabbit polyclonal) | Roche | AB_231 4,622 | IF (1:250) |
| Antibody | Anti-Flag (M2, mouse monoclonal) | Sigma | AB_259529 | IF (1:250) |
| Antibody | Anti-V5 (E10/V4RR), DyLight650 (mouse monoclonal) | ThermoFisher | AB_2537642 | IF (1:100) |
| Antibody | Anti-cut (mouse monoclonal) | DSHB | AB_528186 | IF (1:500) |
| Antibody | Anti-ELAV (rat monoclonal) | DSHB | AB_7E8A10 | IF (1:500) |
| Antibody | Anti-Su(H) (rat polyclonal) | Gift F. Schweisguth | | IF (1:500) |
| Antibody | Anti-Prospero (mouse monoclonal) | DSHB | AB_528440 | IF (1:500) |
| Antibody | Anti-pdm (rabbit polyclonal) | Gift T. Preat | | IF (1:2000) |
| Antibody | Anti-Dl (mouse monoclonal) | DSHB | AB_528194 | IF (1: 500) |
| Antibody | Anti-GFP (B-2, rabbit polyclonal) | Santa Cruz | sc-9996 | IF (1: 500) |
| Antibody | Goat anti-rabbit AlexaFluor 488 | Molecular Probes | AB_2576217 | IF (1: 1000) |
| Antibody | Goat anti-rat AlexaFluor 488 | Molecular Probes | AB_2534074 | IF (1: 1000) |
| Antibody | Goat anti-mouse AlexaFluor 488 | Molecular Probes | AB_2534088 | IF (1: 1000) |
| Antibody | Goat anti-rat AlexaFluor 568 | Molecular Probes | AB_2534121 | IF (1: 1000) |
| Antibody | Goat anti-mouse AlexaFluor 568 | Molecular Probes | AB_144696 | IF (1: 1000) |
| genetic reagent (*D. melanogaster*) | $w^{1118}$ | Bloomington Stock Center | BDSC_3605 | |
| genetic reagent (*D. melanogaster*) | *neurD> GFP* | F.Schweisguth | | |
| genetic reagent (*D. melanogaster*) | $sca^{BP2}$ | Bloomington Stock Center | BDSC_7320 | |
| genetic reagent (*D. melanogaster*) | *UAS sca-RNAi* | Vienna *Drosophila* Resource Center | VDRC#44,527 | |
| genetic reagent (*D. melanogaster*) | *sca::GFSTF* | Bloomington Stock Center | BDSC_64443 | |
| genetic reagent (*D. melanogaster*) | *neur> ph (PLCd)::RFP* | F.Schweisguth | | |
| genetic reagent (*D. melanogaster*) | *UAS-MCFO-1* | Bloomington Stock Center | BDSC_64085 | |
| genetic reagent (*D. melanogaster*) | $UAS-rac1^{N17}$ | Bloomington Stock Center | BDSC_6292 | |
| genetic reagent (*D. melanogaster*) | $Dl^{7}$ | Bloomington Stock Center | BDSC_485 | |
| genetic reagent (*D. melanogaster*) | *UAS-ph::GFP* | Y.Bellaiche | | |
| genetic reagent (*D. melanogaster*) | $neur^{p72}> Gal4$ | Y.Bellaiche | | |

Antibodies and fly lines used.
Fly genotypes corresponding to each figure and video.

| Figures / Videos | | Genotypes |
|---|---|---|
| **Figure 1** | A-C | $neurD> GFP/+; neur^{p72}> Gal4\ tub> Gal80^{ts}/+$ |
| | | |
| **Figure 2** | A-C | $neur> ph\ (PLC\delta)::mRFP/+; pnr> Gal4\ tub> Gal80^{ts}/+$ |
| | | $neur> ph\ (PLC\delta)::mRFP/+; pnr> Gal4\ tub> Gal80^{ts}/UAS-\ rac1^{N17}$ |
| | D | $neur> ph\ (PLC\delta)::mRFP$ |
| | | $neur> ph\ (PLC\delta)::RFP; sca^{BP2}$ |
| | E | $neurD> GFP/+; pnr> Gal4\ tub> Gal80^{ts}/+$ |

*Continued on next page*

*Continued*

| Figures / Videos | | Genotypes |
|---|---|---|
| | | *neurD> GFP/+; pnr> Gal4 tub> Gal80$^{ts}$/UAS- rac1$^{N17}$* |
| | F | *sca::GFSTF/+; neur> ph (PLCδ)::mRFP/+* |
| | | |
| | | |
| **Figure 3** | A-C | *neurD> GFP/+; neur$^{p72}$> Gal4 tub> Gal80$^{ts}$/+* |
| | | *sca$^{BP2}$/sca$^{BP2}$; neur$^{p72}$> Gal4,UAS-H2B::RFP/+* |
| | D, G | *neurD> GFP/+; neur$^{p72}$> Gal4 tub> Gal80$^{ts}$/+* |
| | | *neurD> GFP/+; Dl$^7$/+* |
| | E, G | *neurD> GFP/+; neur$^{p72}$> Gal4 tub> Gal80$^{ts}$/+* |
| | | *sca$^{BP2}$/+; neur$^{p72}$> Gal4,UAS-H2B::RFP/+* |
| | F, G | *neurD> GFP/+; neur$^{p72}$> Gal4 tub> Gal80$^{ts}$/+* |
| | | *sca$^{BP2}$/ neurD > GFP; Dl$^7$/+* |
| | H | *sca::GFSTF/CyO* |
| | | |
| **Figure 4** | A-B | *w$^{1118}$* |
| | | *sca$^{BP2}$/sca$^{BP2}$* |
| | C1 | *y,w, hs-Flp; UAS-ph::GFP/+; pnr> Gal4/ UAS-MCFO1* |
| | C3, D | *y,w, hs-Flp; +/+; pnr> Gal4 tub> Gal80$^{ts}$/UAS-MCFO1* |
| | | *y,w,hs-Flp;UAS-RNAi-sca/+; pnr> Gal4 tub> Gal80$^{ts}$/UAS-MCFO1* |
| | | |
| **Figure 5** | | *y,w,hs-Flp;UAS-RNAi-sca/+; pnr> Gal4 tub> Gal80$^{ts}$/UAS-MCFO1* |
| | | |
| | | |
| **Figure 1—figure supplement 1** | | *neurD> GFP/+; neur$^{p72}$> Gal4 tub> Gal80$^{ts}$/+* |
| | | *sca$^{BP2}$/sca$^{BP2}$; neur$^{p72}$> Gal4,UAS-H2B::RFP/+* |
| | | |
| **Figure 1—figure supplement 2** | | *neurD> GFP/+; neur$^{p72}$> Gal4 tub> Gal80$^{ts}$/+* |
| | | *sca$^{BP2}$/sca$^{BP2}$; neur$^{p72}$> Gal4,UAS-H2B::RFP/+* |
| | | |
| **Figure 2—figure supplement 1** | | *neurD> GFP/+; pnr> Gal4 tub> Gal80$^{ts}$/+* |
| | | *neurD> GFP/+; pnr> Gal4 tub> Gal80$^{ts}$/UAS- rac1$^{N17}$* |
| | | *neurD> GFP/UAS RNAi-sca; pnr> Gal4 tub> Gal80$^{ts}$ /+* |
| | | |
| **Figure 1—figure supplement 3** | A, F | *neurD> GFP/+; pnr> Gal4 tub> Gal80$^{ts}$ /+* |
| | B, F | *neurD> GFP/+; pnr> Gal4 tub> Gal80$^{ts}$/UAS- rac1$^{N17}$* |
| | C, F | *neurD> GFP/UAS RNAi-sca; pnr> Gal4 tub> Gal80$^{ts}$ /+* |
| | D, F | *neurD> GFP/+; neurp72> Gal4 tub> Gal80ts/+* |
| | E, F | *scaBP2/scaBP2; neurp72> Gal4,UAS-H2B::RFP/+* |
| | | |
| **Figure 2—figure supplement 2** | | *sca::GFSTF/CyO* |

*Continued on next page*

*Continued*

| Figures / Videos | | Genotypes |
|---|---|---|
| | | |
| *Figure 3—figure supplement 1* | A, C | sca^BP2/+; neur^p72> Gal4,UAS-H2B::RFP/+ |
| | B, D | sca^BP2/sca^BP2; neur^p72> Gal4,UAS-H2B::RFP/+ |
| | | |
| *Figure 3—figure supplement 2* | A | neurD> GFP/+; neur^p72> Gal4 tub> Gal80^ts/+ |
| | | sca^BP2/sca^BP2; neur^p72> Gal4,UAS-H2B::RFP/+ |
| | B | sca::GFSTF/UAS RNAi-Sca; pnr> Gal4 tub> Gal80 ts /+ |
| | C | neurD> GFP/+; pnr> Gal4 tub> Gal80^ts/+ |
| | | neurD> GFP/UAS RNAi-sca; pnr> Gal4 tub> Gal80^ts /+ |
| | | |
| *Figure 3—figure supplement 3* | A, E | neurD> GFP/+; neur^p72> Gal4 tub> Gal80^ts/+ |
| | B, E | neurD> GFP/+; Dl^7/+ |
| | C, E | sca^BP2/+; neur^p72> Gal4,UAS-H2B::RFP/+ |
| | D, E | sca^BP2/neurD > GFP; Dl^7/+ |
| | | |
| *Figure 4—figure supplement 1* | | UAS-ph::GFP/+; pnr> Gal4 /+ |
| | | UAS-ph::GFP/UAS RNAi-Sca; pnr> Gal4/+ |
| | | |
| | | |
| *Video 1* | | neurD> GFP/+; neur^p72> Gal4 tub> Gal80^ts/+ |
| | | |
| *Figure 2—video 1* | | neur> ph (PLCδ)::mRFP; pnr> Gal4 tub> Gal80^ts/+ |
| | | |
| *Figure 2—video 2* | | neur> ph (PLCδ)::mRFP/+; pnr> Gal4 tub> Gal80^ts/UAS- rac1^N17 |
| | | |
| *Figure 2—video 3* | | sca::GFSTF/+; neur> ph (PLCδ)::mRFP/+ |
| | | |
| *Video 2* | | neurD> GFP/+; neur^p72> Gal4 tub> Gal80^ts/+ |
| | | sca^BP2/sca^BP2; neur^p72> Gal4,UAS-H2B::RFP/+ |
| | | |
| *Figure 5—video 1* | | UAS-RNAi-sca/+; pnr> Gal4 tub> Gal80^ts/UAS-MCFO1 |

## Fly strains

Standard methods were used to maintain fly stocks. The *Gal4/UAS* expression system was used to express several constructions in the mechanosensory bristle lineage as listed in the key resources table. Two drivers were used: *neuralised^p72> Gal4* (**Bellaïche et al., 2001**) expressed in SOPs and their descendants and *pannier> Gal4* (**Sato and Saigo, 2000**) expressed in the dorsal most domain all along pupal stage. For temporal control of transgene expression, *Gal4* drivers were combined with *tub-Gal80^ts*. Crossed flies, developing embryos and larvae were maintained at 18 °C and then pupae were shifted to 30 °C to allow the expression of *Gal4*. For the conditional inactivation of *sca* and the ovexpression of Rac dominant negative form, the shift was done from 12 hr APF to 19 hr APF at the moment of the mitotic wave. To visualise SOP membranes specifically, we used a transgenic line

expressing the pleckstrin homology domain of PLCδ fused to RFP (*ph-RFP*) under the control of the *neur* regulatory sequences *neur> ph(PLCδ)-RFP* (*Couturier et al., 2012*).

## MultiColor FlipOut (MCFO) labelling

MCFO-1 flies (*Nern et al., 2015*) were crossed with *UAS-RNAi-sca, pnr> Gal4, tubGal80ts*. For the control, the crosses were maintained at 18 °C until the progeny reached adulthood. For conditional inactivation of *sca*, pupae were shifted to 30 °C between 12 hr APF to 19 hr APF, then returned to 18 °C until they reached adulthood. In both cases, adults were heat-shocked at 37 °C for 12.5 min (to activate flipase expression), then left for two days at 25 °C to allow the expression of tags. After removing the head, the flies were fixed in 4% paraformaldehyde for 48 hr and rinsed in T-PBS (0.1% Triton X100 in PBS). After dissection, thoracic ventral ganglions were incubated with primary antibodies overnight at 4 °C. After three T-PBS washes, ganglions were incubated with secondary antibodies for 1 hr at room temperature. Ganglions were mounted in Glycerol-PBS (80% glycerol in PBS) and imaged the same day.

## Immunostaining

Dissected nota from pupae at 16 hr APF for Scabrous detection or 24 hr APF for axon labelling were processed as described previously (*Gho et al., 1999*). The antibodies used are described in the key resources table. Incubations with primary antibodies were done overnight at 4 °C and with the secondary antibodies for 1 hr at room temperature. Nota were mounted in Glycerol-PBS (80% glycerol in PBS, 1% propylgalate).

## Confocal microscopy

Immunostaining was observed with an Olympus BX41 fluorescence microscope (objective 40 X/1.30 or 63 X/1.25) equipped with a Yokogawa spinning disc and a CoolSnapHQ2 camera driven by Metaview software (Universal Imaging).

For co-immunodetection of Scabrous and Delta, images were acquired with a Leica SP8 confocal microscope, with the HC PL APO CS2 93 X/1.30 GLYC objective. We tuned the white light laser (WLL) to 650 nm for the excitation. The detector was a HyD. The pixel size was 0.087 μm and the z step size was 0.332 μm. Deconvolution was done with Huygens software. All images were processed with Fiji software (*Schindelin et al., 2012*).

## Time-lapse recording

In vivo imaging was carried out as described previously (*Gho et al., 1999*). The temperature during the recording was controlled using a homemade Peltier device.

## Video and image analysis

The division time of SOPs was analysed for each row with Fiji. The first dividing SOP ($SOP_0$) provided the 'zero time'. For other SOPs, the timing of division was calculated according to this temporal reference. $SOP_0$ also defined the 'rank zero'. The rank of neighbouring SOPs was incremented by one unit from this spatial reference. In order to compare several nota and take account for the neurogenic effect (the increase in the number of SOPs per row) observed in $sca^{BP2}$, we measured the distance between SOPs. In $sca^{BP2}$, the mean number of SOPs was 18 ± 3 separated by a distance of 22 ± 7.2 μm, while in other genetic backgrounds, there are 13 ± 1 SOPs separated by a distance of 30 ± 9 μm. For each graph, the metric rank corresponds to the rank multiplied by the mean distance between two SOPs.

## Heatmaps

SOPs, identified using fly lines with the *neuralised* fluorescent construction, in pupae from 15 hr APF were tracked by live imaging and the time of their division (identified as the anaphase mitotic figures) recorded. The position of cells at the moment of the division as well the relative time of division, encoded according to a rainbow scale, were used to construct each heatmap. The absolute time of division of the first cell which divides in each row was taken as temporal reference. In the dorsal thorax, no migration occurs, so cells keep roughly the same position during the period of live recordings.

## Measurement of protrusion length

Live recording from *neur> ph(PLCδ)-RFP* pupae at 15 hr APF were obtained by combining (max intensity) z-stacks (series of confocal optical sections separated by 1 μm) acquired every 3 min using a 40 X oil immersion objective. For each SOP analysed, the protrusion extension area was calculated from the convex hull of the polygonal line enclosing the extremities of the longer protrusions at each time frame using Fiji software (*Figure 2B*).

## Measurement of axon length

The length of axons in the nota and of the axon terminals in the thoracic ganglion was measured using the Simple Neurite Tracer plugin of Fiji software. To compare the axon lengths of all the nota, the lengths were normalised to the longest axon in each row. The position of the neuron in each row was defined in the same as in the analysis of time of division. To estimate the extent of axon terminal branching in the thoracic ganglion, we measured the length of primary branches between the root and the first branch point (see *Figure 4* C2).

## Cleaning reflex assay

Experiments were performed on headless flies. After decapitation, flies were allowed to recover for 1 hr, then pasted at the tip of needle. The most posterior thoracic bristles in the square formed by the four dorso-central macrochæte and the left dorso-central macrochæte were stimulated by air puffs from an Eppendorf microinjector. Specifically, a 0.1 s puff of 15 hPa was delivered every 0.5 s for a minute. For each fly, the stimulation was done 10 times. The number of puffs required to elicit the first leg movement was counted and the mean for each fly was calculated and plotted.

## Statistics

For the mitotic wave study, all statistical analyses were performed using statistical software R. To compare division time, Linear Mixed Models (LMM) were used. Normality of residuals and homoscedasticity were first assessed. Then LMM was performed using various variables. When a condition was found to be non-significantly different (e.g. Left or Right side of the notum) data were pooled but fly identification was maintained as a fix factor. For example, formula such as this one: lmer(TIME~ COND + POS+ ROW + (1|ID), data = data) was used to assess the impact of TIME according to POS (position) and COND (genetic condition in this example), corrected for ROW and for fly ID as a fixed factor. This test will yield significant influences of factors corrected by the others as well as p-values. Post-hoc analyses were performed for the factor (e.g. ROW) only when the overall model was significant for this factor. For behavioural statistical tests, LMM was used to consider behavioural repetitions using fly ID as a fixed factor and the repetition number. Unpaired *t* tests and Fisher's exact tests were done using Prism seven software.

## Acknowledgements

We thank Marie-Emilie Terret (CIRB, Paris, France) and Bassem Hassam (ICM, Paris, France) for critical and constructive comments on the manuscript. We thank Sophie Gournet for the *Figure 6*. We are grateful Chloé Jean Baptiste Simonne and Julie Perrin for their participation in the experiments presented respectively in *Figure 4C–D* and *Figure 4—figure supplement 1*. This work was funded by institutional support from the Centre National de la Recherche Scientifique (CNRS) and Sorbonne University.

## Additional information

### Funding

| Funder | Grant reference number | Author |
| --- | --- | --- |
| Centre National de la Recherche Scientifique | | Michel Gho |

| Funder | Grant reference number | Author |
|--------|------------------------|--------|

The funders had no role in study design, data collection and interpretation, or the decision to submit the work for publication.

## Author contributions

Jérôme Lacoste, Conceptualization, Data curation, Formal analysis, Investigation, Methodology, Resources, Writing – original draft, Writing – review and editing; Hédi Soula, Formal analysis, Writing – review and editing; Angélique Burg, Resources; Agnès Audibert, Conceptualization, Investigation, Methodology; Pénélope Darnat, Investigation; Michel Gho, Conceptualization, Funding acquisition, Investigation, Methodology, Writing – original draft, Writing – review and editing; Sophie Louvet-Vallée, Conceptualization, Data curation, Investigation, Methodology, Writing – original draft, Writing – review and editing

## Author ORCIDs

Sophie Louvet-Vallée http://orcid.org/0000-0001-7577-2329

## Decision letter and Author response

Decision letter https://doi.org/10.7554/eLife.75746.sa1
Author response https://doi.org/10.7554/eLife.75746.sa2

## Additional files

### Supplementary files

- Appendix 1—figure 2—source code 1. Source code for model description.
- Appendix 1—figure 2—source code 2. Source code for minimal model description.
- Transparent reporting form

### Data availability

All data generated or analysed during this study are included in the manuscript and supporting files.

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

## Appendix 1

### Model Description

#### Phenomenological model

##### Model description and hypotheses

Basically, two alternative strategies may be used to model our observation of coordinated division of SOP cells. On the one hand, we may assume that SOP cells have positive interactions and when entering into mitosis induce mitosis in neighbouring SOP cells. On the other hand, we may assume a negative cell-cell interaction between SOP cells. In this case, SOP cells would prohibit each other from entering into mitosis and cells would be released from this inhibition when their neighbour SOP cells divide. These two strategies may be challenged under experimental conditions where SOP cell-cell interactions would be impaired, for instance after overexpression of rac1N17 or sca mutants. Indeed, we expect that the delay between the mitosis of SOP cells and SOP0 would be increased if cell-cell interactions were positive (mitogenic interactions), and reduced in the alternative condition where cell-cell interactions are negative (anti-mitogenic interactions). Under these same experimental conditions, we observed that SOP cells entered earlier into mitosis. As such, we have favoured a model that assumes an inhibitory nature of the cell-cell interaction (assumptions 1 and 2, see below). Moreover, since we observed that all cells ultimately divide, we assumed (assumptions 3 and 4) that they have an intrinsic timer that triggers division once a certain level of activity is reached. This timer would be slowed down by the interactions with neighbour SOP cells. We show that these simple assumptions allow us to model cell behaviour that approaches what we observe with the actual data. We first describe this model mathematically in abstract terms in the following section and provide a more biologically plausible description that will be able to predict real data in the next section: Activator-Inhibitor model.

##### Simple model

In this section, we describe a very simple model to explain the the shape of the wave. Let's have $k \geq 0$ the cells' indices. Cell's have a waiting time to divide $t_k$ this waiting time is the time when some cell's value $x_k$ reach a constant threshold $\theta$ assumed equal for all cells. Therefore, $t_k$ is the time when $x_k(t_k) = \theta$.

In order to reach $\theta$ we assume that cells have two speeds: one $v$ is when the cell is not inhibited. Starting from 0 a non inhibited cell take $\theta/v$ time to divide. When cells are inhibited the speed is reduced to $\lambda v$ with $0 \leq \lambda \leq 1$. A value $\lambda$ close to 0 is a strong inhibition a value close to one is a light inhibition.

Assuming at first that all cells except cell $k = 0$ are inhibited and that cells inhibit their next neighbor $k + 1$ and are inhibited by their previous neighbor $k - 1$. We can compute the time to division for cell 0 since it is not inhibited: $\theta/v$.

For cell $k$ we have the following relationship:

$$\theta = \lambda v t_{k-1} + v(t_k - t_k - 1)$$

Since the cell $k$ has been inhibited until the cell $k - 1$ has divided, the time for division is the time from the start with neighbor inhibition $\lambda v t_{k-1}$ and then since the cell $k - 1$ does not inhibit anymore after $t_{k-1}$ the remaining time to the threshold is $v\left(t_k - t_{k-1}\right)$ at full speed.

We can rearrange:

$$\theta = \lambda v t_{k-1} + v(t_k - t_k - 1)$$

$$v t_k = \theta - \lambda v t_{k-1} + v t_{k-1}$$

$$t_k = a t_{k-1} + b$$

with $a = 1 - \lambda$ and $b = \theta/v = t_0$. This sequence can be solved exactly:

$$t_k = (1 - \lambda)^k \left(\frac{\theta}{v} - \frac{\theta}{\lambda v}\right) + \frac{\theta}{\lambda v}$$

With $\lambda$ close to 1 no inhibition - we obtain $t_k = b = \theta/v$ every cells divides at the same time.

For low values of $\lambda$ - high inhibition - the curve display a wave of consecutive divisions. At the limit of total inhibition $\lambda = 0$ the division time is a line $t_k = bk = k\theta/v$.

If we take $v = 1$ and $\lambda = 0.5$ we have: $t_0 = 1$

$$t_1 = 1.5$$

$$t_2 = 1.75$$

$$t_3 = 1.875$$

In this example, we have a propagating wave of division that accelerates through time flattening the curve. In case of no inhibition the curve is almost flat passes the first division.

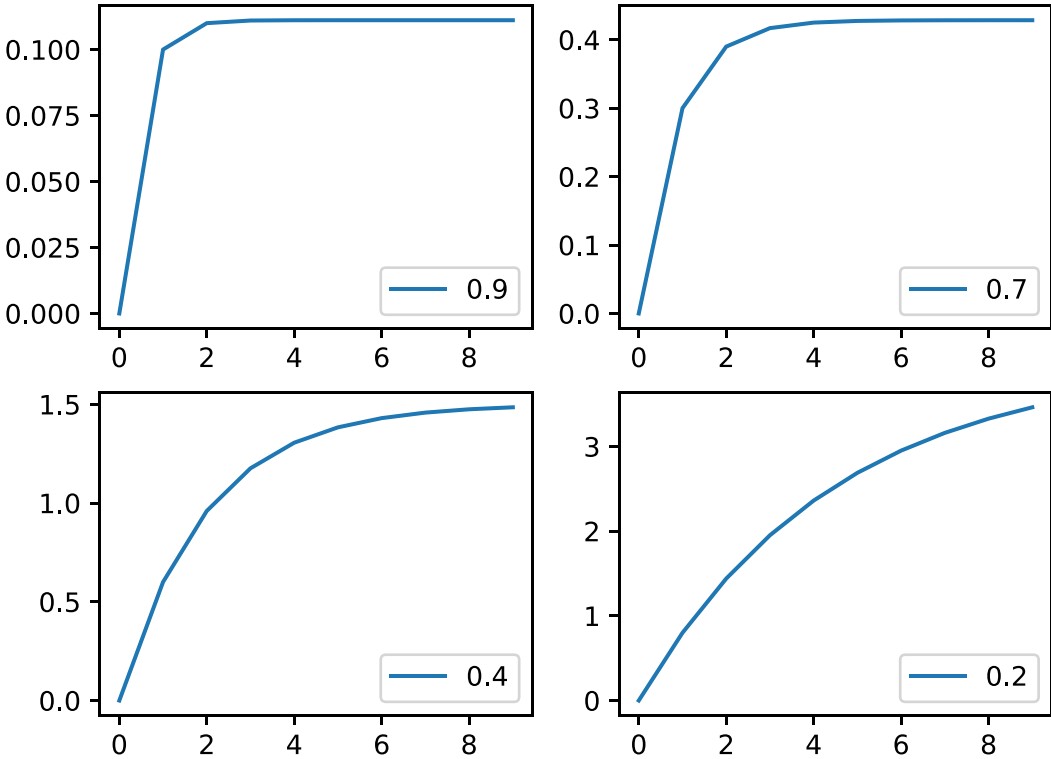

**Appendix 1—figure 1.** Example of the phenomenological model with $\theta = 1$ and $v = 1$ for $k \in \{0, \ldots, 9\}$. For $\lambda \in \{.9, 0.7, 0.4, 0.2\}$ for low to strong inhibition. Please note that the y scales are different.

## Activator-Inhibitor Model
### Model description

In order to compare with the experimental data, we developed another mathematical model. In this model we make similar assumptions: we first assume that (1) neighboring progenitor cells are in a row and inhibit each other to enter into mitosis through cell contacts and (2) this inhibition is switched off when cells divide. We also assume that (3) cell division occurs whenever a certain cellular compound concentration (called $A$) reaches a certain threshold ($\theta$). Finally, (4) this compound is produced at a constant rate but this production is inhibited (decreased) by direct neighbors cells thus preventing the current cell from dividing. Using dynamical systems, this can be modeled as (for a given cell indexed by $k$):

$$\rho \frac{dA_k}{dt} = \frac{r}{1 + I_k} - A_k \tag{1}$$

$$\delta \frac{dI_k}{dt} = \mu \, \mathrm{H}\left(A_{k+1} - \theta\right) + \mu \, \mathrm{H}\left(A_{k-1} - \theta\right) - I_k \tag{2}$$

with

$$H(x) = \begin{cases} 1 & x < 0 \\ 0 & x \geq 0 \end{cases}$$

As mentioned $A$ describes the compound concentration with constant production rate ($r$) inhibited by $I$ another compound that decreases $A$'s total production rate. All degradations are at a constant rate ($\rho$ and $\delta$) as well. Inhibition is computed cell-wise and depends on neighbors' status. The production of inhibition is driven by μ. Higher values of μ indicate greater inhibition from neighboring cells. Neighboring cells whose $A$ is below the threshold (i.e that are not dividing) maintain the inhibition. Whenever the cell divides, inhibition vanishes. Please note that the extent of the inhibition is for direct neighbors only hence the +1 and -1 in the index.

Since $A$ production occurs even through inhibition $A_k(t)$ keeps on increasing and when eventually the inhibition from one side is lifted up it reaches the threshold more rapidly than the previous cells: the speed of the process is inversely proportional to the inhibition. The mechanism described in the simpler model is reproduced the same way in this model.

## Methods

We simulate $2N + 1$ cells from the interval $[-N, N]$ for testing purpose and the exact number of available rows from the division time data for fitting purpose. The system of equation is comprised of $2(2*N+1)$ differential equations that were solved using scipy python package and function odeint (which is a Runge-Kutta of order four solver). Initial conditions were $A_k(0) \sim \mathcal{U}(0, \theta/2)$ for $-N \leq k \leq N$ where $\mathcal{U}$ is the uniform distribution. This excepts $A_0(0) = 2\theta$ (therefore the center cell at $k = 0$ is the first divided cell at $t = 0$). For inhibition, $I_k(0) = \mu$ for $-N \leq k \leq N$. Inhibition from extremal points of the interval is supposed to be on all the time: $I_{-N-1} = I_{N+1} = \mu$ for all $t$. We can assume to have rescale our variables in order to have $\theta = 1$ for all simulations.

*Appendix 1—figure 2 continued on next page*

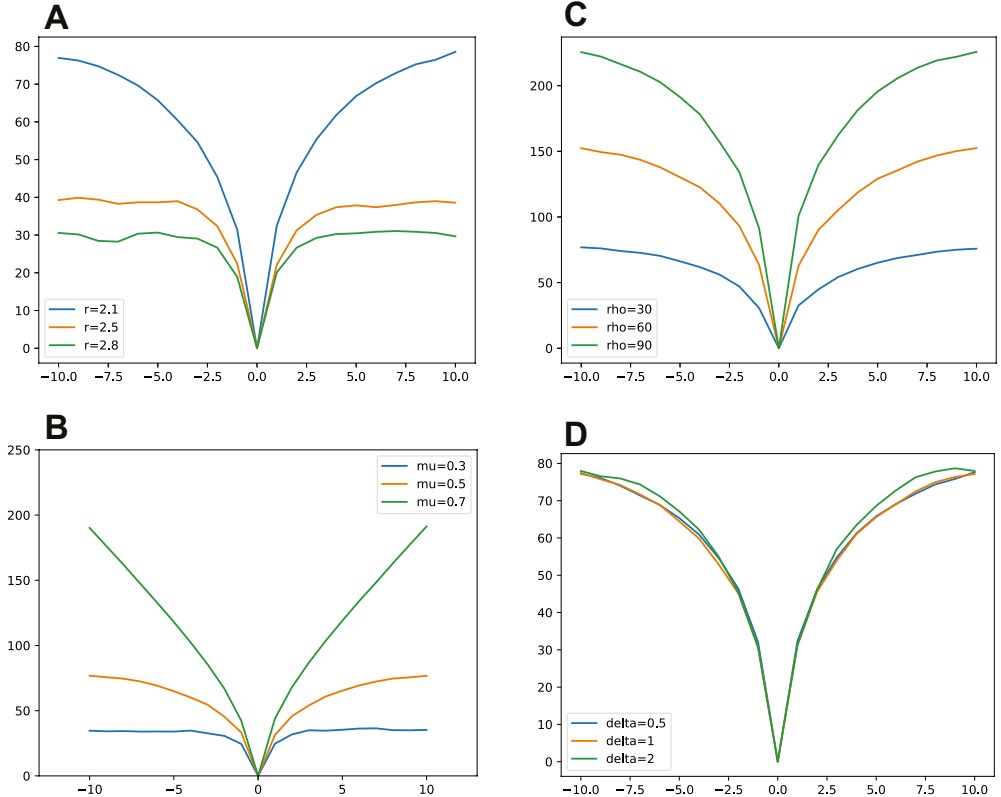

**Appendix 1—figure 2.** TOP LEFT: Simulation different production rate $r \in \{2.1, 2.5, 2.8\}$ (other parameters are $\rho = 30$, $\mu = 0.5$, $\delta = 1$, $\theta = 1$). TOP RIGHT: Simulation with various temporal constant $\rho \in \{30, 60, 90\}$ (other parameters are, $r = 2.1$, $\mu = 0.5$, $\delta = 1$, $\theta = 1$). This adjusts the temporal scale but not the shape of the wave BOTTOM LEFT simulation with various inhibition strength with $\mu \in \{0.5, 0.6, 0.7\}$ (other parameters are, $r = 2.1$, $\rho = 30$, $\delta = 1$, $\theta = 1$). Strong inhibition here (0.7) imposes a more linear division time shape whereas low inhibition allows for quicker division with neighboring cells dividing almost simultaneously. BOTTOM RIGHT: simulation with various inhibition time constant with $\delta \in \{10, 0.5, 1\}$ (other parameters are, $r = 2.1$, $\rho = 30$, $\mu = 0.5$, $\theta = 1$).

The online version of this article includes the following source code for appendix 1—figure 2:

- **Appendix 1—figure 2—source code 1.** Source code for model description.
- **Appendix 1—figure 2—source code 2.** Source code for minimal model description.

   This figure shows the main features of the model: to obtain similar division rates with successive cell divisions (a linear increasing division time function) a strong inhibition is needed. On the other hand, almost simultaneous cells division (without any wave) is obtained by decreasing the amount of inhibition.

## Parameters estimation

To estimate parameters, we perform $L_2$ norm minimization:

$$\chi^2(r, \rho, \mu, \delta) = \Sigma_{k=0}^{N} \left( T_k - t_k(r, \rho, \mu, \delta) \right)^2 \tag{3}$$

   $T_k$ are the division time for position $k \in \{\dots, -3, -2, -1, 0, 1, 2, 3, \dots\}$ from the data and $t_k(r, \rho, \mu, \delta)$ is the division time from the model with parameters $r, \rho, \mu, \delta$. We performed Nelder-Mead parameters optimisations in order to reach minimal $\chi^2$.

   In this model, all parameters cannot be identified exactly at once since the only data fitted is the division time: having four parameters has produced redundant information to fit the division time from the data. To avoid problems of identification, we assumed that production rate $r$ is unlikely to change. We assumed also that degradation for the inhibitory compound was higher therefore

their half-life constant much lower ($\delta$ is small compared to $\rho$) and also set it to a constant. These assumptions allows to reduce the problem to two free parameters - μ and $\rho$ - to perform the fitting. In this case, decreasing the number of free parameters had no significant impact in $L_2$ minimisation as fits were comparable while maintaining stable results.

In order to obtain noisy representations, the algorithm performs 10 simulation of the differential equation with random starting values for $A$ (uniform distribution between 0 and $\theta/2$) except for the cell at position 0 which is set above threshold. The times of division are computed for each cell and the average is used as values $t_k$.

