## [Editor Report]

This study investigates development of the mechanosensory organ on *Drosophila* notum. It combines live imaging, mathematical modelling, genetics and behavioural analysis to show that, in the peripheral nervous system of *Drosophila*, entry of progenitor cells into mitosis is spatially and temporally controlled. The authors suggest that this ensures proper targeting of sensory neurons within the ventral nerve cord. This timing is important for axonogenesis and proper spatial arborization, ultimately influencing the animal's behaviour. The study will be of broad interest to those who work on the developmental of sense organs, and in general on the role of timing in development.

---

## [Decision Letter]

**Decision letter after peer review:**

Thank you for submitting your article "A neural progenitor mitotic wave is required for asynchronous axon outgrowth and morphology" for consideration by *eLife*. Your article has been reviewed by 3 peer reviewers, including Sonia Sen as Reviewing Editor and Reviewer #1, and the evaluation has been overseen by Claude Desplan as the Senior Editor. The following individual involved in review of your submission has agreed to reveal their identity: Tatsumi Hirata (Reviewer #2).

Essential revisions:

This study investigates development of the mechanosensory organ on *Drosophila* notum using various genetic techniques. They combine live imaging, mathematical modeling, genetics and behavioral analysis to show that in the peripheral nervous system of *Drosophila*, entry of progenitor cells into mitosis is spatially and temporally controlled. This, the authors suggest, ensures proper targeting of sensory neurons within the ventral nerve cord. The study will be of broad interest to those who work on developmental processes, and particularly to those interested in sense organ development.

While appreciating the quality of the work and its presentation, we have a few suggestions for the authors that they should be able to address within the time frame of a revision:

1. Related to the model: There are some questions and concerns about the model that have been raised in two of the reviews. They are largely clarification issues. Could the authors please respond to each of them in their revised version? The individual points can be found in the detailed reviews below.

2. All three reviewers raised concerns about possible effect the genetic perturbations might have on bristle development in general. This would have implications for the axogenesis phenotypes that the authors report. So, could the authors check that this is not the case? For example, they could put the directly driven neur-H2B::RFP in their genetic background and look at the characteristic arrangement and size of the bristle lineage nuclei; or stain it with markers such as pros, cut, elav, or any others of their choice that report cell fate identities within an SOP lineage.

3. Finally, there are concerns that in some cases, for example in the of timing of the mitotic wave and its effect on axogenesis, the interpretations of the data are too strongly made. The authors should revisit the text in these sections (Please read the individual reviews for the detailed comments.)

*Reviewer #1 (Recommendations for the authors):*

We found the study to be interesting, thoroughly executed, well documented, and a pleasure to read. We have a few comments and concerns, which I'm sure the authors should be able to address.

Related to the model

– Is there any interaction between the rows? If SOPs are extending filopodia in all directions, one would expect the wave to move concentrically, and not in the A-P axis.

– One of the predictions from this would be that the level of the inhibitor (Sca) should also follow a wave like progression – start off high in a G2 arrested SOP and drop off as the SOP divides. From the expression patterns shown, this is not possible to see, and the live imaging of Sca:GFP shows only one or two SOPs. Could the authors please comment on this?

– If all SOPs are creating both the activator and the inhibitor, why does self-inhibition not occur?

– The authors mention an acceleration of division with time. We could not see this clearly in the experimental data, but also don't understand what, in the model, might result in it. While section 1 in the Model Description attempts to explain this acceleration with a simpler example, in case of the activator-inhibitor model the "acceleration" behaviour seems to depend on the threshold theta. Specifically, whether theta is below r/(1+2*mu) or between r/(1+2*mu) and r/(1+mu) seems to dictate whether the wave will have an acceleration or an asymptotically constant velocity respectively. Could the authors please verify/comment on this?

– Could the authors also elaborate on how they solved the equations (numerical scheme, initial conditions, boundary conditions) in the model description supplement? Specifically in the activator-inhibitor model, what were the boundary conditions used? The value of theta seems to be unmentioned in the figures both in the main article and the model description. This might be useful for others who want to take a similar approach in other systems.

– If inhibition stops upon losing contact, then the inhibition from cell k to its neighbours, and the inhibition from the neighbours to cell k, should all go to zero. This effect is better captured by a multiplication of Heaviside theta functions i.e. H(A_{K^+^1} – theta) * H(A_{k} – theta) + H(A_{k-1} – theta) * H(A_{k} – theta). Making this change allowed us to reproduce the figures in the article while the original equations did not yield the same results. Could the authors please comment on this?

Related to axon targeting and behaviour

– We were concerned that some of the phenotypes might be due to inappropriate cell fate specification in the SOP due to the manipulation of the notch pathway. Can the authors show that this is not affected?

General

– In the introduction, could the authors please talk about SOP cell division and fates, and the role of the notch pathway in it? This will help the reader interpret their results in the context of the entire process.

*Reviewer #2 (Recommendations for the authors):*

1) This is a hypothesis-driven paper. Individual phenotypes caused by gene manipulations look coherent only based on several hypotheses provided by the authors, but each hypothesis is not well supported by previous or present studies. Even though this reviewer generally agrees that original speculation and interpretation should be more highly appreciated and included in papers, and also that the hypotheses in this manuscript serve as a good guide for readers to digest the contents, some parts of the manuscript are just too assertive. Hypothetical arguments should be addressed as hypothetical. A few details are explained below.

2) The live analysis of mitosis timing is the strength of this study. Although the mitotic wave is not very clear in the heat-mapped pictures, the quantified data and the simple mathematical model help convincing understanding of the wave. The model intuitively explains the difference of characters in the wave propagation from the origin to second cells and that among other further SOPs. Although the authors discuss that the mitotic origin is determined by other mechanism, I still do not understand why the second origin does not appear upon suppression of the inhibitory signal. Or does it in fact appear? Notch-signaling mutants generally have more bristles in the thorax. Does this phenotype actually influence the analysis of scabrous mutants? The authors should discuss them in the paper.

3) In scabrous mutants, the mitotic wave is disrupted, but is not abandoned or synchronized. The word choice should be more careful. The critical caveat of the scabrous mutants is that positions of mitotic origin are not the same as those in the wild-type (Figure S2). This means that the mutation somehow affected cell-intrinsic characters of SOP cells themselves. Therefore, the mutant phenotypes described later cannot be attributed solely to changes of mitotic timing. Does the RNAi mutant show a similar distribution of the origins?

4) The cytoplasmic protrusions may be a candidate of cell-cell-communication, but this hypothesis still lacks experimental support in the SOP system. Because Rac1 is pleiotropic, it is similarly possible that the dominant negative form affected other aspects of the cells. The authors should be clearer about the limitations. Rephrase the conclusive subtitle of this paragraph.

5) The association between neuron generation timing and axon patterns has no supportive evidence at this moment. It is a pure speculation with no ground. The paper by Usui-Ishihara and Simoson referred in this manuscript examines inter-raw but not intra-raw differences. They used the length of bristles as an indirect scale of differentiation timing but did not examine the birth timing of the cells per se. Top of that, these authors only discussed the rough correlation in a few sentences in the paper. In this regard, this manuscript's claim that axonal patterning is altered because of the change of mitotic timing is obviously overstatement and misleading to readers.

*Reviewer #3 (Recommendations for the authors):*

There are a number of points where the clarity and presentation of the manuscript could be improved:

1. In Figure 1C, the authors use a linear regression to fit the data from individual SOP row, while in 1D the combined data from all three row is fitted with their model. However, no explanation is given for the difference in fit (linear vs non-linear) in the individual versus combined data. This should briefly be clarified in the text and/or theory supplement.

2. The data motivating the assumptions used to build the model should be better introduced. On line 02-to-06, the authors state these assumptions, but without further explanations the choice of model seems rather arbitrary. In order not to break the flow of the manuscript, the authors could refer in the results to a more detailed section in the methods where they explain the thinking behind their model and cite the relevant literature.

3. In general, it would be helpful to the reader to write the GAL4 driver used for the experiments in the actual Figures (e.g. 2A, 5B…).

4. Figure 4C, D: what developmental stage is this? It should be clearly indicated in the legend or main text.

5. For the experiments where tubG80ts is used to temporally control sca-RNAi and dominant-negative Rac, it would be useful to state the timings in the Figure legends or indicate it on the figures themselves. These details are important to exclude confounding effects due to interference with other aspects of bristle development than the mitotic wave.

---

## [Author Response]

Reviewer #1 (Recommendations for the authors):We found the study to be interesting, thoroughly executed, well documented, and a pleasure to read. We have a few comments and concerns, which I'm sure the authors should be able to address.Related to the model– Is there any interaction between the rows? If SOPs are extending filopodia in all directions, one would expect the wave to move concentrically, and not in the A-P axis.

This is an interesting point. SOP cells extend protrusions in all directions and they can connect SOP cells from adjacent rows. However, the rows have different developmental timing as indicated in the introduction:

“In the dorsal region of the thorax (notum), SOPs appear sequentially from five parallel proneural rows: first rows 5, followed by rows 1 and 3 and finally rows 2 and 4 (Usui and Kimura, 1993; Corson et al., 2017).”

As a consequence, we assumed that SOP in one row can not affect the mitotic timing of SOPs of adjacent rows. By these reasons, we have developed our model considering only intra-row cell-cell interactions.

– One of the predictions from this would be that the level of the inhibitor (Sca) should also follow a wave like progression – start off high in a G2 arrested SOP and drop off as the SOP divides. From the expression patterns shown, this is not possible to see, and the live imaging of Sca:GFP shows only one or two SOPs. Could the authors please comment on this?

In the epithelium, we failed to observe a reduction in the global Sca expression following SOP cell division. We think that protrusions allow a focal Sca mediated signalization with a dynamics that is not possible to reveal by live imagining. In order to clarify this point in the text we have modify the sentence:

“It is plausible that Sca, transported through protrusions, is required to maintain SOPs in G2 arrest, though we cannot formally rule out the possibility that Rac1^N17^ overexpression affects Sca secretion *per se*.“

by:

“Since, we do not observe a reduction of the global level of *sca* expression associated with the wave progression, it is plausible that Sca, required to maintain SOPs in G2 arrest, is delivered focally through protrusions that are difficult to follow with in vivo analysis. Although we favored this possibility, we cannot formally rule out the possibility that Rac1^N17^ overexpression affects Sca secretion *per se* without affects *sca* expression”

– If all SOPs are creating both the activator and the inhibitor, why does self-inhibition not occur?

This is a very interesting point raised by the reviewer. Our data show that the inhibitory activity of Sca requires protrusions. And since we have never observed that these protrusions make contact with the cell from which they originate, it is unlikely that an auto-inhibitory mechanism is taking place.

– The authors mention an acceleration of division with time. We could not see this clearly in the experimental data, but also don't understand what, in the model, might result in it. While section 1 in the Model Description attempts to explain this acceleration with a simpler example, in case of the activator-inhibitor model the "acceleration" behaviour seems to depend on the threshold theta. Specifically, whether theta is below r/(1+2*mu) or between r/(1+2*mu) and r/(1+mu) seems to dictate whether the wave will have an acceleration or an asymptotically constant velocity respectively. Could the authors please verify/comment on this?

The main reason for the acceleration is that, according to our model, the division occurs whenever a certain compound reaches a threshold this compound being produced as a certain rate. When inhibited the compound production rate is slowed but when inhibition is lifted this compound concentration is closer to the threshold due precisely to the previous albeit decreased production, therefore crossing it more rapidly than the previous cell since it had more time to increase. The activator-inhibitor displays the same behaviour: submitted to the full inhibition the main production rate is r/(1+2mu) when one inhibition is lifted the production rate is r/(1+mu) (which is > r/(1+2mu)). Please note that r/(1+2*mu) is a rate of increase of A (as well as r/(1+mu)) so in order to eventually have a division we need to have always r/(1+mu) > theta (which is always the case).

If however r/(1+2*mu) < theta – very strong inhibition – it’s a special case where division can never happen without an initial cell division.

Indeed: using rho dA/dt = r/(1+2*mu) – A this equation has solution A(t)=(A0 – r/(1+2*mu)) exp(-rho t) + r/(1+2*mu) so A(t) -> r/(1+2*mu) when t tends to infinity – so the limit for A is exactly r/(1+2*mu) when the cell is constantly inhibited.

However, we force a cell division at t=0 so in practice we are never in the case where the production rate is r/(1+2*mu) for all the cells at the start of the simulation. Therefore, all the cells will eventually divide regardless of r/(1+2*mu) being greater or not than theta. Finally, please note that according to our fit in all cases r/(1+2*mu) > theta. And as mentioned the value of mu will dictate the rate of acceleration: the stronger the inhibition the lower the acceleration.

– Could the authors also elaborate on how they solved the equations (numerical scheme, initial conditions, boundary conditions) in the model description supplement? Specifically in the activator-inhibitor model, what were the boundary conditions used? The value of theta seems to be unmentioned in the figures both in the main article and the model description. This might be useful for others who want to take a similar approach in other systems.

We provide this details in the supplement now. Please note also that the code used for the article is provided as well. To answer here: initial conditions are for A is uniformly distributed between 0 and theta/2 for I it is set at mu (halfway between full inhibition and no inhibition).

All cells start as inhibited except for the centred one. For the boundaries cells the inhibition from the outer edges is always on. The scheme is an adaptive Runge-Kutta 4 from the python scipy package function odeint.

– If inhibition stops upon losing contact, then the inhibition from cell k to its neighbours, and the inhibition from the neighbours to cell k, should all go to zero. This effect is better captured by a multiplication of Heaviside theta functions i.e. H(A_{K^+^1} – theta) * H(A_{k} – theta) + H(A_{k-1} – theta) * H(A_{k} – theta). Making this change allowed us to reproduce the figures in the article while the original equations did not yield the same results. Could the authors please comment on this?

Indeed, in the current model, inhibition comes from undivided cells which provide inhibition unto cells even though they have already divided. Therefore, we do not stop inhibition from undivided neighbouring cells when a cell has divided. However we fail to see the difference using referee’s equation: Assuming cell k-1 has divided but not k or K^+^1 : the new rate of A_k (at cell k) will be r/(1+mu) – instead of r/(1+2*mu) and remain thus until cell k has divided in that case the rate will be r – instead of r/(1+mu) in our model until K^+^1 divides where it will be r as well. But this will not affect the division time of k since the equation is different only after it has divided. We have provided the code in order to allow reproduction and comparison.

Related to axon targeting and behaviour– We were concerned that some of the phenotypes might be due to inappropriate cell fate specification in the SOP due to the manipulation of the notch pathway. Can the authors show that this is not affected?

We thank the reviewer for pointing out the necessity of this control experiment. We have included the data showing that cell fate of sensory organ cells is not affected in *sca^BP2^* mutant in Figure S6.

General– In the introduction, could the authors please talk about SOP cell division and fates, and the role of the notch pathway in it? This will help the reader interpret their results in the context of the entire process.

We have added a sentence in the introduction describing the asymmetric division of SOP, the fates of terminal cells and the role of Notch pathway.

“Then, at 16.5 h APF, SOPs resume the cell cycle and divide to generate a posterior secondary precursor cell (pIIa) and an anterior secondary precursor cell (pIIb). After subsequent asymmetric cell divisions, they give rise to the external (the socket and the shaft cells) and internal (the neuron, the sheath and the glial cell) cells respectively (Gho et al., 1999; Fichelson, 2003). During these asymmetric cell divisions, daughter cells acquire different cell fates relied with a differential activation of the Notch (N) pathway (Schweisguth, 2015).”

Reviewer #2 (Recommendations for the authors):1) This is a hypothesis-driven paper. Individual phenotypes caused by gene manipulations look coherent only based on several hypotheses provided by the authors, but each hypothesis is not well supported by previous or present studies. Even though this reviewer generally agrees that original speculation and interpretation should be more highly appreciated and included in papers, and also that the hypotheses in this manuscript serve as a good guide for readers to digest the contents, some parts of the manuscript are just too assertive. Hypothetical arguments should be addressed as hypothetical. A few details are explained below.

In this modified version, we have taken special care to clearly differentiate hypotheses from interpretations or conclusions of our experimental data. We hope that this version will fully address the criticisms raised by the reviewer.

2) The live analysis of mitosis timing is the strength of this study. Although the mitotic wave is not very clear in the heat-mapped pictures, the quantified data and the simple mathematical model help convincing understanding of the wave. The model intuitively explains the difference of characters in the wave propagation from the origin to second cells and that among other further SOPs. Although the authors discuss that the mitotic origin is determined by other mechanism, I still do not understand why the second origin does not appear upon suppression of the inhibitory signal. Or does it in fact appear?

In the model, the SOP_0_ is used as a spatial reference wherever it is physically located in the notum. As such, the model does not modelized ectopic/secondary SOP_0_ localizations.

Notch-signaling mutants generally have more bristles in the thorax. Does this phenotype actually influence the analysis of scabrous mutants? The authors should discuss them in the paper.

We are aware that the neurogenic effect may influence the analysis of *sca* mutant data. For this reason, we have taken in account this observed neurogenetic effect using the metric rank as a representation of SOP location. This strategy allows us both to pool together data from different nota and to compare data obtained under different conditions. The metric rank representation is described in Methods, section “Video and images analysis”, as follows:

“In order to compare several nota and take account for the neurogenic effect (the increase in the number of SOPs per row) observed in scaBP2, we measured the distance between SOPs. In scaBP2, the mean number of SOPs was 18 ± 3 separated by a distance of 22 ± 7.2 μm, while in other genetic backgrounds, there are 13 ± 1 SOPs separated by a distance of 30 ± 9 μm. For each graph, the metric rank corresponds to the rank multiplied by the mean distance between two SOPs”.

3) In scabrous mutants, the mitotic wave is disrupted, but is not abandoned or synchronized. The word choice should be more careful.

We have corrected the text as the reviewer suggested.

The critical caveat of the scabrous mutants is that positions of mitotic origin are not the same as those in the wild-type (Figure S2). This means that the mutation somehow affected cell-intrinsic characters of SOP cells themselves. Therefore, the mutant phenotypes described later cannot be attributed solely to changes of mitotic timing. Does the RNAi mutant show a similar distribution of the origins?

In the *RNAi sca* mutants, we also observe a spread of SOP_0_ localization. We have added a supplementary figure (Figure S2) with this data. Formally, we can imagine that *sca^BP2^* mutation induces an intrinsic change in the SOP properties that would contribute to the SOP_0_ spread. However, this spread of SOP_0_ was also observed in the *rac1^N17^* overexpression context. As such, the more plausible explanation is that the appearance of ectopic SOP_0_ is due exclusively to a reduction in inhibition and not to an intrinsic effect of *sca^BP2^* mutation.

4) The cytoplasmic protrusions may be a candidate of cell-cell-communication, but this hypothesis still lacks experimental support in the SOP system. Because Rac1 is pleiotropic, it is similarly possible that the dominant negative form affected other aspects of the cells. The authors should be clearer about the limitations. Rephrase the conclusive subtitle of this paragraph.

Since Rac1 promotes F-actin polymerization, we are agreed that we cannot exclude that the observed effect on mitotic wave is due to a more general effect of actin polymerization.

We have rephrased the subtitle as:

Rac1^N17^ affects both cytoplasmic protrusions length and SOP mitotic wave progression

5) The association between neuron generation timing and axon patterns has no supportive evidence at this moment. It is a pure speculation with no ground. The paper by Usui-Ishihara and Simoson referred in this manuscript examines inter-raw but not intra-raw differences. They used the length of bristles as an indirect scale of differentiation timing but did not examine the birth timing of the cells per se. Top of that, these authors only discussed the rough correlation in a few sentences in the paper. In this regard, this manuscript's claim that axonal patterning is altered because of the change of mitotic timing is obviously overstatement and misleading to readers.

We agree that the paper by Usui-Ishihara and Simpson examined the inter-raw pattern of microchaetes. However, these authors have written a section entitled: “Central projection patterns of microchaetes in *D. melanogaster*: complexity of branching is dependent on time of birth of precursor cells”. In this section, they mentioned that: “These data suggest that axons of early-born microchaetes have the most complex branching patterns and arborize over a greater area, while those born later have simpler patterns.” And they concluded in the discussion:” Axons of earlier born microchaetes, wherever they were positioned, displayed more branches and arborized over a greater area”. As such, we don’t have the impression to have overstated the data of Usui-Ishihara and Simpson.

In order to avoid any misleading to readers, we have modified the conclusion of this Results section as follow:

“As such, our results led us to hypothesize that the order of birth of neurons in a given row contributes to the future connectivity of these neurons. Further experiments aimed to analyse the axon terminal of sensory organs at a given row are required to confirm this hypothesis.”

We have also modified in the introduction the sentence where we addressed the results of Usui-Ishihara and Simpson. We have replaced the sentence:

“Thus, the earlier the microchætes developed, wherever they were positioned, the more branched they were, arborizing over a greater area within the neuropil”

By

“Thus, the microchætes form the earlier born rows were more branched and arborized over a greater area within the neuropil than those from later born rows”.

Reviewer #3 (Recommendations for the authors):There are a number of points where the clarity and presentation of the manuscript could be improved:1. In Figure 1C, the authors use a linear regression to fit the data from individual SOP row, while in 1D the combined data from all three row is fitted with their model. However, no explanation is given for the difference in fit (linear vs non-linear) in the individual versus combined data. This should briefly be clarified in the text and/or theory supplement.

We have used a linear regression in order to have a simple quantitative characterization of each curve using the mean wave rate as a parameter. In the text, we have explained:

“To characterize each curve, the rate of the mitotic wave was calculated as the mean absolute value of the inverse of the slopes of the linear approximation of the curves for all conditions “.

However, conscient that this is only a roughly characterization of each curve, we have fitted the real data with the model and the parameters obtained are showed in figure S4. We hope that these explanations will clarify the use of these two fits.

2. The data motivating the assumptions used to build the model should be better introduced. On line 02-to-06, the authors state these assumptions, but without further explanations the choice of model seems rather arbitrary. In order not to break the flow of the manuscript, the authors could refer in the results to a more detailed section in the methods where they explain the thinking behind their model and cite the relevant literature.

We have modified the introduction in the model section to better justify the assumptions used to build the model, as follows:

1.1. Model description and hypotheses. Basically, two alternative strategies may be used to model our observation of coordinated division of SOP cells. On the one hand, we may assume that SOP cells have positive interactions and when entering into mitosis induce mitosis in neighbouring SOP cells. On the other hand, we may assume a negative cell-cell interaction between SOP cells. In this case, SOP cells would prohibit each other from entering into mitosis and cells would be released from this inhibition when their neighbour SOP cells divide. These two strategies may be challenged under experimental conditions where SOP cell-cell interactions would be impaired, for instance after overexpression of Rac1^N17^ or *sca* mutants. Indeed, we expect that the delay between the mitosis of SOP cells and SOP0 would be increased if cell-cell interactions were positive (mitogenic interactions), and reduced in the alternative condition where cell-cell interactions are negative (anti- mitogenic interactions). Under these same experimental conditions, we observed that SOP cells entered earlier into mitosis. As such, we have favoured a model that assumes an inhibitory nature of the cell-cell interaction (assumptions 1 and 2, see below). Moreover, since we observed that all cells ultimately divide, we assumed (assumptions 3 and 4) that they have an intrinsic timer that triggers division once a certain level of activity is reached. This timer would be slowed down by the interactions with neighbour SOP cells. We show that these simple assumptions allow us to model cell behaviour that approaches what we observe with the actual data. We first describe this model mathematically in abstract terms in the following section and provide a more biologically plausible description that will be able to predict real data in the next section: Activator-Inhibitor model.

3. In general, it would be helpful to the reader to write the GAL4 driver used for the experiments in the actual Figures (e.g. 2A, 5B…).

We have amended all the figures legends to include the *Gal4* driver as suggested by the reviewer.

4. Figure 4C, D: what developmental stage is this? It should be clearly indicated in the legend or main text.

We apologize for forgetting to specify the developmental stage studied and we thank the reviewer for this remark. We have added the stage (adult) in the figure legend.

5. For the experiments where tubG80ts is used to temporally control sca-RNAi and dominant-negative Rac, it would be useful to state the timings in the Figure legends or indicate it on the figures themselves. These details are important to exclude confounding effects due to interference with other aspects of bristle development than the mitotic wave.

We have added in all figures legends concerned this sentence to precise the experimental strategy of conditional inactivation using *tubGal80^ts^*: “*UAS-tubGal80^ts^* was used to conditionally express *Gal4* by maintaining flies at 18 °C and shifted them to 30 °C from 12 h to 19 h APF.